# LocalV: Exploiting Information Locality for IP-level Verilog Generation

## Abstract

The generation of Register-Transfer Level (RTL) code is a crucial yet labor-intensive step in digital hardware design, traditionally requiring engineers to manually translate complex specifications into thousands of lines of synthesizable Hardware Description Language (HDL) code. While Large Language Models (LLMs) have shown promise in automating this process, existing approaches—including fine-tuned domain-specific models and advanced agent-based systems—struggle to scale to industrial IP-level design tasks. We identify three key challenges: (1) handling long, highly detailed documents, where critical interface constraints become buried in unrelated submodule descriptions; (2) generating long RTL code, where both syntactic and semantic correctness degrade sharply with increasing output length; and (3) navigating the complex debugging cycles required for functional verification through simulation and waveform analysis. To overcome these challenges, we propose *LocalV*, a multi-agent framework that leverages the inherent *information locality* in modular hardware design. LocalV decomposes the long-document to long-code generation problem into a set of short-document, short-code tasks, enabling scalable generation and debugging. Specifically, LocalV integrates hierarchical document partitioning, task planning, localized code generation, interface-consistent merging, and AST-guided locality-aware debugging. Experiments on REALBENCH demonstrate that LocalV substantially outperforms state-of-the-art (SOTA) LLMs and agents, showing the potential of generating Verilog for IP-level RTL design.

## 1 Introduction

The generation of Register-Transfer Level (RTL) code is a core step in digital hardware design. This process is notoriously labor-intensive and error-prone, as engineers must manually translate natural language specifications into thousands of lines of synthesizable Hardware Description Language (HDL) code (e.g., Verilog, VHDL). The promise of Large Language Models (LLMs) to automate this step has spurred rapid innovation. Initial efforts focused on benchmarking general-purpose models (Liu et al., 2023b; Thakur et al., 2023) and developing domain-specific solutions through fine-tuning or data augmentation (Liu et al., 2024c; Cui et al., 2024; Liu et al., 2024b; Zhao et al., 2025). More recently, the field has shifted towards sophisticated agent-based systems that mimic human design workflows. These agents, such as VerilogCoder (Ho et al., 2025) and MAGE (Zhao et al., 2024), decompose complex problems and can operate autonomously or in a human-in-the-loop fashion, as explored in collaborative design platforms like ChatCPU (Wang et al., 2024) and Spec2RTL-Agent (Yu et al., 2025).

Despite strong results on academic benchmarks like VerilogEval (Liu et al., 2023b), a clear gap appears when applying current LLM-based methods to industrial hardware design. This is particularly evident with REALBENCH (Jin et al., 2025), an IP-level benchmark derived from real-world open-source IP, which features significantly longer documentation (197.3 vs. 5.7) and code lengths (241.2 vs. 15.8) compared to VerilogEval. Directly using SOTA models or agents often leads to a sharp drop in performance, with many outputs failing to be even syntactically correct, let alone functionally valid. This gap highlights a mismatch between current model capabilities and the high requirements of real-world hardware engineering. We observe three main challenges:

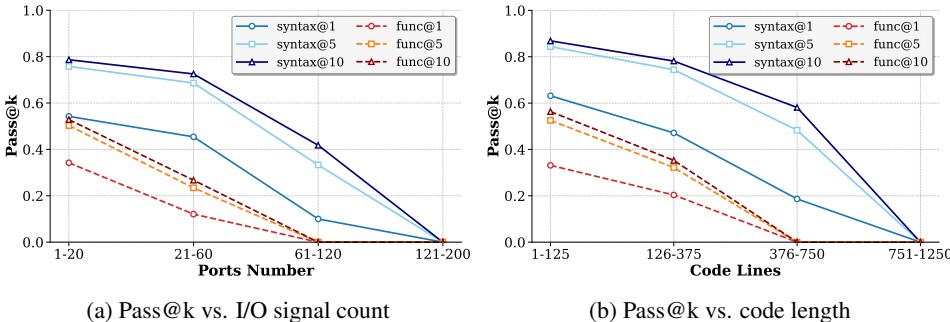

(a) Pass@k vs. I/O signal count

(b) Pass@k vs. code length

Figure 1: Performance of Claude 3.7 Sonnet on REALBENCH: Pass@k vs. (a) I/O signal count and (b) code length (lines), reporting syntactic and functional Pass@k. Accuracy decreases with interface complexity and output length.

**Long-Document Handling.** IP-level hardware specifications are typically verbose and detailed, largely due to the increasing number of I/O signals and submodules. Although modern LLMs support context windows of 32k tokens or more, their ability to generate functionally correct RTL code diminishes as document complexity grows. The accumulation of signal and module details overwhelms the model's limited understanding of hardware semantics. As a result, critical interface constraints are often obscured by irrelevant details, leading to phantom signals, port list mismatches, and logically incorrect Verilog code. This trend is illustrated in Figure 1a, which shows a consistent decrease in LLM accuracy as the number of I/O signals increases.

**Long-Code Generation.** IP-level designs usually involve substantially longer code, which exacerbates the challenges LLMs face in HDL generation—a domain where they already underperform. As shown in Figure 1b, both syntactic and semantic accuracy drop significantly with code length. When the code exceeds 750 lines, even repeated sampling (e.g., 10 times) fails to yield a syntactically valid result. Typical errors include incorrect macro references, use of non-synthesizable constructs, and fundamental syntax errors, underscoring the model's inherent limitations in generating reliable RTL code.

**Complex Debugging Process.** In practice, IP-level hardware verification relies on carefully constructed testbenches to ensure the design conforms to specifications. Each simulation failure triggers a laborious debugging cycle: engineers analyze waveforms to identify faulty signals, trace errors back to ambiguous or misinterpreted specification segments, and iteratively refine the design. This process not only corrects the code but also clarifies ambiguities in the specification itself, using waveform behavior as a definitive reference for refinement.

To address these challenges, we propose LocalV, a multi-agent framework explicitly designed for the real-world IP-level "long-document, long-code" hardware generation problem. Our key observation is that IP-level specifications inherit strong **information locality** from modular hardware design: code fragments can often be generated correctly by relying on only a portion of the document. This suggests that long-document to long-code generation can be decomposed into a set of short-document to short-code tasks without information loss, thereby mitigating the core challenges.

Specifically, LocalV organizes the following workflow as shown in Figure 2: (1) Preprocessing. Documents are partitioned into fragments with hierarchical indices. (2) Planning. Code structure is planned as sub-tasks with assigned document fragments. (3) Generation. Coding agents execute "short-document, short-code" generation for each sub-task. (4) Merging. Fragments are merged into a complete design with interface consistency. (5) Debugging. Error messages and AST-guided waveform analysis trace failures back to specification fragments for locality-aware debugging.

Our contributions are summarized as follows: (1) We realized three fundamental challenges in generating IP-level Verilog code, namely, long-document handling, long-code generation, and the complex debugging process. (2) We observed the information locality principle for IP-level Verilog generation. Based on it, we introduce an index-driven document partitioning mechanism, a fragment-based generation strategy that decomposes complex tasks into manageable subtasks, and a traceable debugging pipeline that maps errors back to relevant specification fragments via AST-guided analysis. (3) Based on these techniques, we present LocalV, a multi-agent framework for

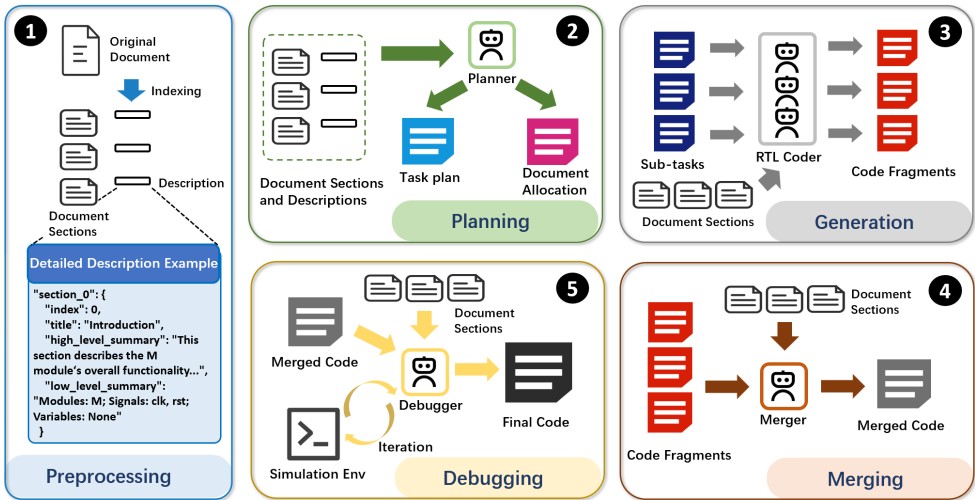

Figure 2: Workflow overview of LocalV.

generating correct Verilog from **IP-level specifications**. LocalV achieves a 45.0% pass rate on RE-ALBENCH (Jin et al., 2025), surpassing SOTA agent-based frameworks by 23.4%.

## 2    RELATED WORK

**LLM-based RTL Generation.**    The application of LLMs to automate RTL code generation has emerged as a promising research area in electronic design automation (EDA). Early explorations (Nair et al., 2023; Chang et al., 2023; Blocklove et al., 2023) focused on evaluating the capability of general-purpose LLMs to translate natural language specifications into Hardware Description Languages (HDLs) like Verilog and VHDL. Foundational benchmarks such as VerilogEval (Liu et al., 2023b) and RTLLM (Lu et al., 2024) were established to systematically assess model performance, revealing both the potential and limitations of off-the-shelf models. To improve performance, subsequent research has focused on domain-specific adaptation through fine-tuning on curated datasets (Liu et al., 2024c; Thakur et al., 2024; Liu et al., 2023a) or optimization via reinforcement learning (Pei et al., 2024; Zhu et al., 2025; Chen et al., 2025). While these models show strong results on well-defined, smaller-scale problems, their effectiveness on real-world, IP-level specifications is fundamentally limited. For many fine-tuned models, this stems from smaller model scales, the lack of training data for IP-level hardware design, and constrained context windows that fail to fully capture complex design documents. More critically, even for large-scale models with extensive context capabilities, the single-pass generation paradigm is ill-suited for the complexity of IP-level design. Attempting to synthesize functionally correct code from a verbose specification in a single attempt struggles to capture the intricate dependencies and hierarchical nature of hardware, often leading to subtle but critical errors.

**Agent-based Frameworks for Hardware Design.**    To overcome the limitations of single-pass generation, the field is shifting towards multi-agent frameworks that emulate the collaborative and iterative nature of human design and verification workflows. This paradigm moves beyond a single monolithic model to a team of specialized agents, each assigned a distinct role. For instance, MAGE (Zhao et al., 2024) explicitly creates a four-agent team responsible for RTL generation, testbench creation, functional evaluation (judging), and debugging, establishing a clear, recursive loop of proposing and refining the design. Similarly, RTLSquad (Wang et al., 2025) organizes its agents into "squads" dedicated to distinct project phases—exploration, implementation, and verification—thereby mimicking the structure of a human engineering team. Central to these systems is a task decomposition phase, where a high-level specification is broken down into a structured plan with manageable sub-tasks. These plans guide the execution of agents focused on coding, planning, and reflection, as seen in frameworks like Spec2RTL-Agent (Yu et al., 2025) and VerilogCoder (Ho et al., 2025). However, this decomposition process faces a critical challenge: translating the orig-

inal specification into intermediate instructions can introduce cascading ambiguity, distorting the design intent. Consequently, debugging becomes severely hampered, as agents must trace errors through these distorted interpretations rather than the source document. Our approach addresses this by maintaining a direct link between the specification and code, grounding the entire process in the original document.

# 3 METHODOLOGY

We begin by formalizing the problem of IP-level Verilog generation (§3.1). We then introduce our core hypothesis, the *information locality* in IP-level hardware specifications, with a quantitative analysis (§3.2). We then present the detailed LocalV pipeline built on this insight(§3.3).

## 3.1 PROBLEM FORMULATION

Our objective is to synthesize a complete Verilog module from a natural language specification. We formally define the problem as follows:

**Input:** A natural language specification document $\mathcal{D}$, represented as an ordered sequence of $N$ semantic textual units (e.g., paragraphs or sections), $\mathcal{D} = \{d_1, d_2, \ldots, d_N\}$. Also, a target module name $m$ and a simulation environment $E$ that provides golden execution feedback (including error messages and behavioral mismatches) for debugging purposes is given.

**Output:** A synthesizable Verilog module $\mathcal{V}_m$. We model the generated code not as a monolithic text file, but as a structured set of $M$ semantic code units, $\mathcal{V}_m = \{c_1, c_2, \ldots, c_M\}$. A code unit $c_j$ represents a functionally cohesive and syntactically complete block of RTL code, such as a module or a statement. The final output file is the concatenation of these units.

**Objective:** The generated module $\mathcal{V}_m$ must be functionally correct and can pass a suite of simulation tests from $E$ against a golden reference testbench, ensuring functional equivalence.

## 3.2 INFORMATION LOCALITY

Our approach is grounded in a core assumption we term **Information Locality**: for any semantic code unit $c_j \in \mathcal{V}_m$, the information required to generate $c_j$ is primarily concentrated within a subset of the specification $\mathcal{D}$. This locality arises directly from the hierarchical and modular nature of hardware design. Complex systems are built from well-defined submodules (e.g., ALUs, register files), and IP-level specifications explicitly mirror this structure: dedicated sections describe each module's behavior, I/O, and internal logic. This creates a natural alignment, where the implementation of a code unit $c_j$ depends predominantly on its corresponding documentation segment. In contrast, general-purpose software specifications often describe high-level algorithms that do not decompose neatly into code-level constructs, leading to more diffuse information sources.

The validity of this locality hypothesis is key. If it holds—as we will quantitatively demonstrate below (Figure 3)—it enables a powerful divide-and-conquer strategy. The quantitative results confirm that the "long-document to long-code" mapping problem can be effectively decomposed into a set of parallelizable "short-document to short-code" subproblems without information loss, dramatically improving tractability.

We quantify information locality by measuring the entropy of the information source distribution for each code unit. Our analysis begins by segmenting the specification $\mathcal{D}$ into paragraphs $\{d_i\}_{i=1}^N$ and the Verilog code $\mathcal{V}_m$ into statements $\{c_j\}_{j=1}^M$. For each code statement $c_j$, we compute its semantic similarity $\text{sim}(d_i, c_j)$ (cosine similarity of Qwen3-Embedding-0.6B embeddings) to every specification paragraph $d_i$ and transform them into conditional probability distribution $P(d_i \mid c_j)$ using a softmax function with temperature $\tau = 0.1$:

$$P(d_i \mid c_j) = \frac{\exp(\text{sim}(d_i, c_j)/\tau)}{\sum_{k=1}^N \exp(\text{sim}(d_k, c_j)/\tau)}. \tag{1}$$

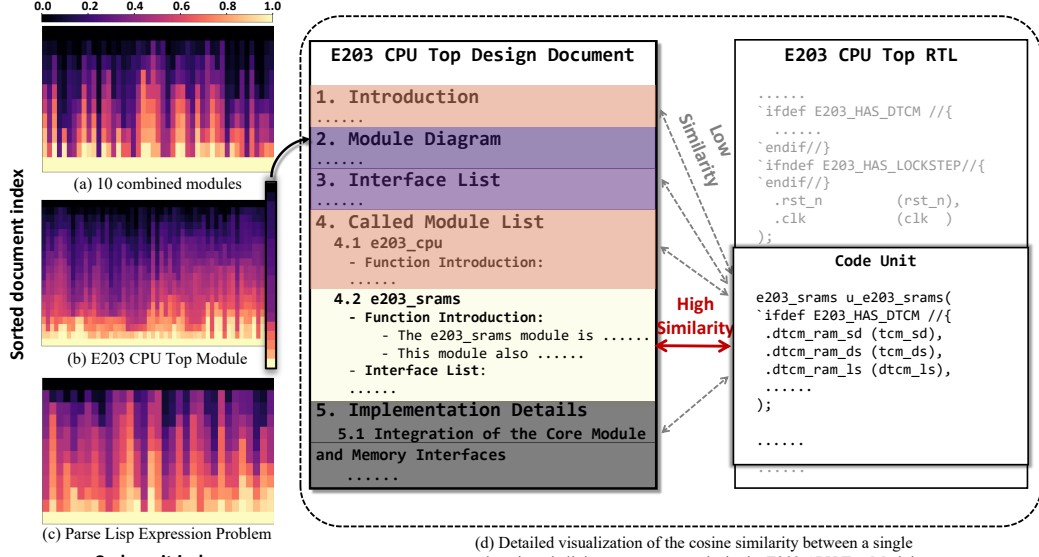

Figure 3: Heatmaps of normalized cosine similarity across three tasks. Each column represents a code unit and its sorted cosine similarity to all document paragraphs. Values in each column are independently normalized to the range [0, 1], where lower values indicate higher information locality. (a) 10 randomly selected and then combined modules from VerilogEval, demonstrating extremely high information locality (since they are totally independent of each other) with $\bar{H}_{\text{norm}} = 0.8206$. (b) The E203 CPU Top Module from REALBENCH, showing high information locality with $\bar{H}_{\text{norm}} = 0.8680$. (c) The Parse Lisp Expression problem, a typical software task, with $\bar{H}_{\text{norm}} = 0.9126$. (d) A detailed visualization of the cosine similarity between a single code unit and all document paragraphs in the E203 CPU Top Module.

The locality for $c_j$ is then assessed by the entropy of this distribution:

$$H(c_j) = -\sum_{i=1}^{N} P(d_i \mid c_j) \log_2 P(d_i \mid c_j), \tag{2}$$

where lower entropy indicates that information is concentrated in a small number of textual units, thus supporting the locality hypothesis.

To ensure comparability across specifications of different lengths, we normalize the entropy by its theoretical maximum, $H_{\max} = \log_2 N$, which occurs under a uniform distribution. The normalized entropy for a code unit is:

$$H_{\text{norm}}(c_j) = \frac{H(c_j)}{\log_2 N}. \tag{3}$$

This yields a scale-invariant measure where $H_{\text{norm}}(c_j) \in [0, 1]$. To report a single locality score for an entire design, we average the normalized entropy across all $M$ code units:

$$\bar{H}_{\text{norm}} = \frac{1}{M} \sum_{j=1}^{M} H_{\text{norm}}(c_j). \tag{4}$$

A lower $\bar{H}_{\text{norm}}$ indicates stronger overall locality, and this metric is comparable across experiments with varying $N$ and $M$.

We evaluate three settings to contrast information locality: (a) a **synthetic Verilog benchmark** (10 concatenated VerilogEval cases) as an ideal locality baseline (lower bound); (b) the **hardware IP** e203_cpu_top from REALBENCH; and (c) a **software counterpart** (LeetCode "Parse Lisp Expression" in Python) with comparable length. Row-normalized heatmaps and the average normalized entropy $\bar{H}_{\text{norm}}$ quantify locality strength. As shown in Figure 3, the hardware design (b) exhibits strong locality ($\bar{H}_{\text{norm}} = 0.8680$), much closer to the ideal (a) ($\bar{H}_{\text{norm}} = 0.8206$) than the software case (c) ($\bar{H}_{\text{norm}} = 0.9126$). This pattern holds across REALBENCH, where the average

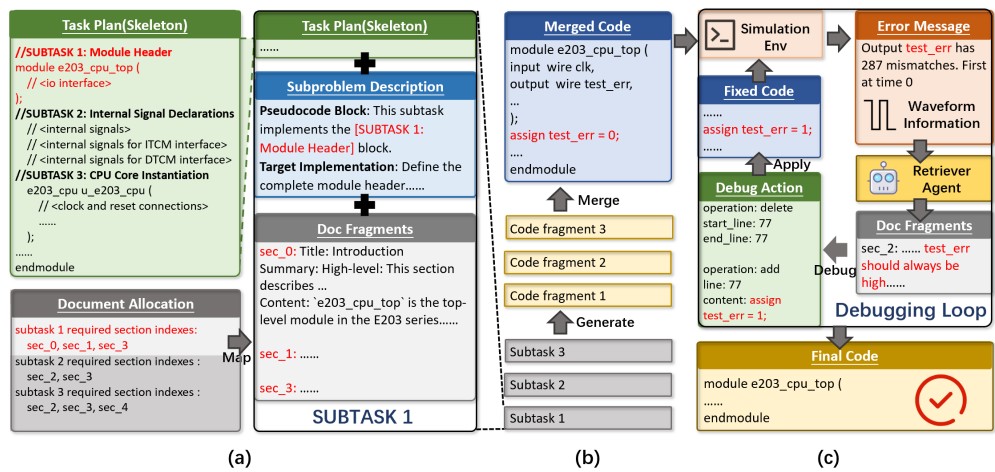

Figure 4: The detailed workflow of LocalV. (a) Output of the planning stage, illustrating the structure of a sub-task. (b) Overview of the code generation and merging process. (c) Overview of the debugging loop and the generation of the final code.

$\bar{H}_{\mathrm{norm}} = 0.8406$ further confirms stronger locality in hardware specifications. To validate the robustness of these findings, we further computed $\bar{H}_{\mathrm{norm}}$ using diverse similarity models, including BM25, Qwen3-Embedding-8B, and DeepRTL2 (Llama) (Liu et al., 2025). The detailed comparisons are presented in Table 1.

Table 1: $\bar{H}_{\mathrm{norm}}$ of Different Similarity Method

| Module | BM25 | DeepRTL2 | Qwen-0.6B | Qwen-8B |
|---|---|---|---|---|
| 10 Combined Modules | 0.0669 | 0.9255 | 0.8206 | 0.8167 |
| RealBench Average | 0.2083 | 0.9231 | 0.8406 | 0.8607 |
| E203 CPU Top Module | 0.4036 | 0.9504 | 0.8680 | 0.8873 |
| Parse Lisp Expression | 0.5412 | 0.9699 | 0.9126 | 0.9453 |

### 3.3 LocalV Overview

We now introduce our novel multi-agent framework, **LocalV**, designed to automate the generation of Verilog code from long natural language documentation. The overall workflow is depicted in Figure 4 and detailed in the subsequent sections.

### 3.3.1 Preprocessing

The first stage of our pipeline structures the input documentation for efficient retrieval and comprehension by the agents. Given a raw design document, we split the text into coherent paragraphs. For each paragraph, an LLM is prompted to generate a dual-level description that indexes the source content:

**Semantic level:** Provides a high-level summary of the paragraph's functional intent, such as "interface specification for the DMA controller" or "timing constraints for the DDR memory interface." This supports agents who require a conceptual understanding of a module's purpose.

**Lexical level:** Extracts fine-grained hardware-specific entities—including signal names, module identifiers, macros, and parameters—to ensure precise retrieval of low-level details that may be omitted in semantic summaries.

The resulting description serves as keys indexing the original text segments and is used in subsequent stages of the pipeline.

### 3.3.2 PLANNING AND TASK DECOMPOSITION

Upon receiving the indexed documentation from the previous stage, the **Planner Agent** constructs the overall structure of the final Verilog code and generates a corresponding skeleton. This skeleton is expressed as pseudo-code containing syntactic placeholders that represent various code components—such as submodule instantiations or signal assignments.

The agent then decomposes the skeleton into sub-tasks, each corresponding to a code fragment that requires implementation. For every sub-task, the **Retriever** queries the hierarchical index to identify and retrieve the most relevant document sections, attaching them as focused context. This targeted contextualization not only narrows the scope of each generation step but also ensures alignment with the original specification.

Unlike approaches that naively partition hardware into submodules or create intermediate representations, our fragment-based decomposition introduces no additional complexity. All sub-tasks contribute directly to the same global design, each addressing a well-defined portion of the code. This method maintains tight alignment with the final output and mitigates common issues such as objective drift that may arise from self-generated intermediate goals.

### 3.3.3 RTL GENERATION

With the sub-tasks and their associated documentation contexts prepared, multiple instances of the **RTL Agent** proceed to fill the placeholders in the code skeleton. Each agent is assigned a specific sub-task and operates within a constrained context, allowing it to focus exclusively on its local objective. This narrow focus facilitates an accurate translation of the specification into synthesizable Verilog for the corresponding code segment, thereby reducing errors such as phantom signals and enhancing the overall quality of the generated code fragments.

### 3.3.4 CODE FRAGMENTS MERGING

After all **RTL Agents** complete fragment generation, the **Merge Agent** integrates the fragments into a correct Verilog module. To resolve potential inconsistencies or implementation errors that may arise during merging, the **Retriever Agent** first fetches relevant sections from the original documentation. Using this retrieved context, the **Merge Agent** then refines and integrates the fragments using this additional information together with the generated code, ensuring that the final output is correct and coherent.

### 3.3.5 LOCALITY-AWARE DEBUGGING

LocalV's debugging pipeline leverages **information locality** to efficiently trace errors back to their relevant documentation segments. The process begins by curating error messages from the simulation environment to extract key signals—such as syntax error locations or functional mismatches, and root-cause signal information from waveform analysis (inspired by VerilogCoder's (Ho et al., 2025)). Crucially, the **Retriever Agent** then uses this error context to fetch the small subset of documentation fragments that are locally relevant to the faulty code section, as determined by the underlying information locality hypothesis. A dedicated **Debug Agent** subsequently synthesizes this focused context—the error details and the retrieved documentation—to produce precise, line-number-aware edit actions (e.g., inserting or deleting specific lines). This debug loop iterates until the code is error-free or a predefined iteration limit is reached, efficiently minimizing corrective overhead by avoiding reprocessing the entire specification.

## 4 EXPERIMENTS

We evaluate LocalV's performance on realistic hardware design tasks through a series of experiments. We first describe our experimental setup, then report the main results comparing LocalV against baselines, and finally perform an ablation study to quantify the contribution of components.

Table 2: Syntax and functional pass rate comparison on the REALBENCH benchmark.

| Method | SDC | | AES | | E203 CPU | | ALL | |
|---|---|---|---|---|---|---|---|---|
| | Syn. | Func. | Syn. | Func. | Syn. | Func. | Syn. | Func. |
| *Model Baselines* | | | | | | | | |
| Claude-3.7 | 41.4% | 11.7% | 46.6% | 31.6% | 42.7% | 20.6% | 42.8% | 19.6% |
| DeepSeek-V3 | 44.2% | 15.3% | 55.8% | 23.3% | 19.5% | 7.5% | 28.9% | 10.9% |
| DeepSeek-R1 | 28.5% | 7.1% | 66.6% | **50.0%** | 12.5% | 10.0% | 21.6% | 13.3% |
| Qwen3-32B | 25.3% | 15.3% | 32.4% | 16.6% | 8.3% | 6.2% | 14.7% | 9.4% |
| GPT-4o | 14.2% | 0.0% | 50.0% | 16.6% | 5.0% | 0.0% | 11.6% | 1.6% |
| GPT-5 | 7.1% | 0.0% | 50.0% | 33.3% | 30.0% | 20.0% | 26.6% | 16.6% |
| *Agent Baselines* | | | | | | | | |
| MAGE (Claude) | 57.1% | 21.4% | 66.6% | 33.3% | 62.5% | 20.0% | 61.6% | 21.6% |
| VerilogCoder (Claude) | 0.0% | 0.0% | 0.0% | 0.0% | 0.0% | 0.0% | 0.0% | 0.0% |
| **LocalV (DeepSeek-V3)** | **64.2%** | **28.5%** | **50.0%** | **50.0%** | **60.0%** | **35.0%** | **60.0%** | **35.0%** |
| **LocalV (Claude)** | **78.5%** | **35.7%** | **83.3%** | **50.0%** | **72.5%** | **47.5%** | **75.0%** | **45.0%** |

## 4.1 SETTINGS

**Benchmarks.** We adopt REALBENCH (Jin et al., 2025), a challenging benchmark specifically designed for real-world, IP-level Verilog generation. REALBENCH comprises 60 RTL generation tasks drawn from three IPs: AES encoder/decoder cores (6 modules), an SD card controller (14 modules), and a CPU core (40 modules). REALBENCH emphasizes practical applicability through long-form natural language specifications (averaging 10k tokens) and substantial implementation complexity (approximately 320 lines of Verilog code per target module on average). To further assess the generalization capabilities of LocalV, we also incorporate the non-agentic part of the cid003 (spec-to-rtl) subset from the CVDP (Pinckney et al., 2025) benchmark. We exclude the agentic part since it requires abilities other than RTL generation, such as reading and writing files via the command line, and navigating, organizing, and pinpointing issues across multiple code files. These requirements are beyond the topic of LocalV (IP-level spec-to-rtl), and are orthogonal to LocalV's agent ability.

**Metrics.** We evaluate models on syntactic and functional correctness using each benchmark's predefined testbenches, and report both syntax and functional pass rate as the metric. We use Pass@1 in Table 2 4, and extend to Pass@k (OpenAI, 2021; Liu et al., 2023b) in some analysis. As reported in Table 2, the pass rates for direct prompting model baselines are averaged over 20 independent generations per task, whereas Agent baselines are evaluated using a single generation per task.

**Baselines.** We establish comprehensive baselines comprising both standalone models and agent-based systems. For standalone models, we evaluate both commercial and open-source models, including Claude (Claude-3.7-sonnet-250219) (Anthropic, 2025), DeepSeek-V3 (DeepSeek-v3-250324) (Liu et al., 2024a), DeepSeek-R1 (DeepSeek-r1-250528) (Guo et al., 2025), Qwen3-32B (Yang et al., 2025), GPT-4o (OpenAI, 2024), and GPT-5 (OpenAI, 2025). For agent-based approaches, we compare against SOTA methods, MAGE (Zhao et al., 2024) and VerilogCoder (Ho et al., 2025), both implemented using Claude-3.7-sonnet-250219. Our LocalV method is evaluated on two different backbone models: Claude-3.7-sonnet-250219 and DeepSeek-v3-250324.

## 4.2 MAIN RESULTS

Table 2 presents the evaluation results on the challenging REALBENCH benchmark. This benchmark proves particularly difficult for current LLMs, as evidenced by the modest 19.0% functional Pass@1 achieved even by the strong Claude-3.7-sonnet-250219 model. Notably, our LocalV (Claude) surpasses the base model's Pass@20 performance (35.0%) with just a single generation, highlighting its significant advantages for IP-level hardware design tasks.

When compared against agent baselines, LocalV demonstrates superior performance over both MAGE (overall 23.4%) and VerilogCoder. It is important to note that MAGE typically relies on extensive high-temperature sampling to generate candidate programs—a computationally expensive approach for long-form code generation. To ensure a fair comparison with LocalV's single-shot

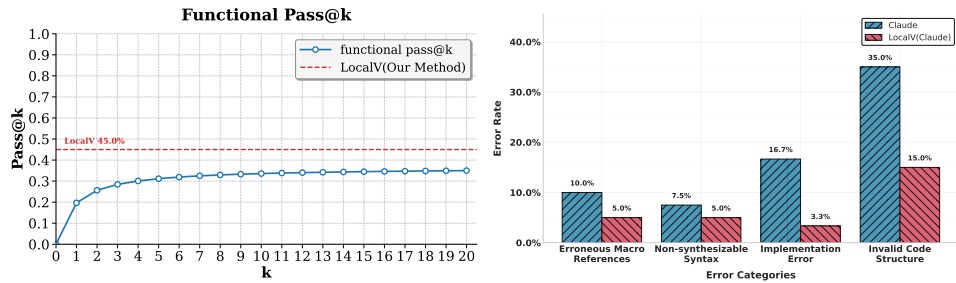

(a) Sampling results of Claude 3.7 Sonnet vs. LocalV.

(b) Distribution of syntactic error types for Claude 3.7 Sonnet and LocalV.

Figure 5: Comparison between LocalV and direct sampling

setting, we limited MAGE's candidate size to two and allocated an equivalent debugging iteration budget. VerilogCoder employs a ReAct-style workflow (Yao et al., 2023) that performs well on simpler tasks but struggles with IP-level complexity. Without specific design adaptations for complex hardware generation, its per-agent success rates diminish as context length increases, and its nondeterministic orchestration leads to high computational costs and low completion rates. Under reasonable cost constraints, VerilogCoder failed to solve any REALBENCH instances.

In contrast, LocalV achieves stronger performance with substantially improved resource efficiency, enabled by its streamlined agent architecture and precise task decomposition strategy. The method generates each code fragment only once, performs a single merge operation, and executes a bounded debugging schedule (maximum 10 iterations), producing high-quality solutions while maintaining controlled generation costs.

Also, we present a comparison between LocalV and direct sampling using Claude in Figure 5. To demonstrate the superior functional accuracy of LocalV, we plot its performance alongside the Pass@k values of direct sampling in Figure 5a. The results indicate that the Pass@k of direct sampling tends to converge after $k = 10$, yet remains substantially lower than the accuracy achieved by LocalV. Furthermore, we provide a detailed breakdown of syntax error categories for both methods in Figure 5b. The results show that LocalV consistently exhibits a lower syntax error rate across all categories, highlighting its robust syntactic performance in diverse problem settings.

Table 3: Functional pass rate comparison on the cid003.

| Method | Func. |
| --- | --- |
| Claude-3.7 | 48.72% |
| MAGE (Claude) | 44.87% |
| **LocalV (Claude)** | **61.50%** |

We also evaluate our method on the cid003 subset of CVDP benchmark to show its effectiveness on short tasks. Following the setting of CVDP, we use pass@1 as the functional accuracy. As shown in Table 3, while these tasks have relatively short contexts (avg. ∼1,100 tokens) and thus do not fully demonstrate LocalV's long-context ability, LocalV still significantly outperforms direct sampling and MAGE, showing strong robustness. These results indicate that LocalV maintains superior performance across tasks with different specification styles, supporting its scalability and robustness beyond REALBENCH.

## 4.3 ABLATION STUDIES

We conduct ablation studies on LocalV (based on Claude) in Table 4.

First, replacing indexed document fragments with the full specification significantly degrades performance across all benchmarks. The hierarchical indexing mechanism proves essential for managing IP-level complexity, as long specifications introduce substantial irrelevant content that distracts the model and harms generation quality. Even with other components intact, removing indexing alone causes a notable 10.0% drop in overall functional pass rate.

Table 4: Ablation study on the REALBENCH benchmark.

| Method | SDC | | AES | | E203 CPU | | ALL | |
|---|---|---|---|---|---|---|---|---|
| | Syn. | Func. | Syn. | Func. | Syn. | Func. | Syn. | Func. |
| **LocalV** | **78.5%** | **35.7%** | **83.3%** | **50.0%** | **72.5%** | **47.5%** | **75.0%** | **45.0%** |
| **w/o index** | 64.2% | 21.4% | 100.0% | **50.0%** | 57.5% | 37.5% | 63.3% | 35.0% |
| **w/o index & debug** | 35.7% | 7.1% | 50.0% | 33.3% | 57.5% | 22.5% | 51.6% | 20.0% |
| **w/o index & debug & plan** | 35.7% | 7.1% | 50.0% | 33.3% | 45.0% | 22.5% | 43.3% | 20.0% |

Second, the debugging component demonstrates the importance of code correctness. When both indexing and debugging are removed, performance drops to 20.0%—only marginally above the base model's Pass@1. This indicates that while our task decomposition strategy addresses core challenges, the debugging stage is vital for ensuring functional correctness of the generated IP blocks.

Finally, the planner provides complementary benefits by enhancing syntactic correctness and orchestrating the generation process. While its impact on functional accuracy is less pronounced than indexing and debugging, it contributes to syntactic accuracy, and the full ablation (without index, planner, and debug) yields the lowest performance (20.0%), confirming the planner's role in maintaining structural coherence for complex hardware design tasks.

Overall, these results confirm that information locality is the unifying principle behind LocalV's effectiveness. The hierarchical indexing establishes locality by focusing on relevant document fragments, while the planner maintains locality through structured generation. The debugging component extends this approach by tracing errors to specific documentation segments for targeted corrections. The performance degradation when compromising locality—whether through fragmented generation without indexing or monolithic generation without planning—demonstrates that locality-aware decomposition is essential for IP-level code generation under constrained budgets.

## 5 CONCLUSION

We present LOCALV, a multi-agent framework with a workflow tailored to IP-level hardware design. Our study observes and validates the information locality of IP-level hardware specifications. Most RTL fragments can be correctly implemented based on a partial specification. Building on this insight, we design a novel hierarchical indexing strategy, a fragment-oriented task decomposition, and a locality-aware debugging loop. In REALBENCH, a real-world IP-level benchmark, LocalV delivers a 10% improvement, advancing the practical generation of reliable RTL code with LLM.

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

## A  THE SYSTEM LEVEL RESULT OF REALBENCH

Figure 6 presents the design hierarchy of RealBench and the corresponding performance of LocalV. Specifically, it details the verification outcomes for (a) an SD card controller, (b) an AES encoder/decoder core, and (c) the Hummingbirdv2 E203 CPU Core. A "Pass" denotes successful module generation by LocalV, whereas a "Fail" indicates an unsuccessful attempt. The hierarchical tree structure within the figure visually represents the intricate task interdependencies in RealBench, underscoring its inherent complexity.

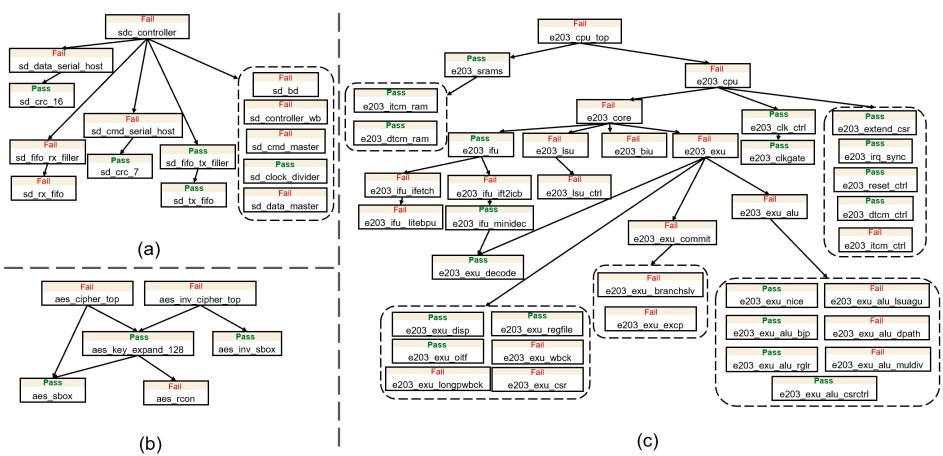

Figure 6: The system level result of RealBench.

## B  INTERMEDIATE RESULTS OF LOCALV

To better illustrate LocalV's workflow, this section delves into the detailed intermediate results for the **e203_exu** problem in REALBENCH. We'll display the outputs generated by LocalV agents, including document fragments, pseudocode, plans, code fragments, and debug actions, providing a comprehensive understanding of the process.

---

**Document Fragments**

"section_0": "The 'e203_exu' module represents the execution unit (EXU) of a processor core, responsible for executing instructions received from the instruction fetch unit (IFU) and handling exceptions, pipeline flushes, and write-back operations. This module integrates submodules for decoding, dispatching, ALU operations, commit handling, and more, ensuring proper execution of instructions and system state management.",

"section_1": "| Direction | Port Name | Width | Description |
| ——— | ———— | ——— | —————————————————— |
| output | commit_mret | 1 | Indicates that an MRET instruction has been committed. |
| output | commit_trap | 1 | Indicates that a trap (e.g., exception) has been committed. |
| output | exu_active | 1 | Indicates whether the execution unit is currently active. |
| output | excp_active | 1 | Indicates whether an exception is currently active. |
| output | core_wfi | 1 | Indicates that the core is in a waiting-for-interrupt (WFI) state. |
| output | tm_stop | 1 | Indicates that the timer should stop. |
| output | itcm_nohold | 1 | Indicates no hold on the instruction TCM. |
| output | core_cgstop | 1 | Core clock gating stop signal. |
| output | tcm_cgstop | 1 | TCM clock gating stop signal. |
| input | core_mhartid | E203_HART_ID_W | Hardware thread ID of the core. |
| input | dbg_irq_r | 1 | Debug interrupt request signal. |
......",
......

---

```
Pseudocode
// === SUBTASK 1: Module Header and IO Port Definitions ===
module e203_exu (
// <basic interface signals>
// <debug control interface signals>
// <IFU IR stage interface signals>
// <flush interface signals>
// <LSU write-back interface signals>
// <AGU ICB interface signals>
// <optional CSR_NICE interface signals>
// <optional NICE interface signals>
);
// === SUBTASK 2: Internal Signal Declarations ===
// <register file interface signals>
// <decode stage signals>
// <dispatch stage signals>
// <OITF signals>
// <ALU signals>
// <commit stage signals>
// <CSR signals>
// <long-pipeline write-back signals>
// <write-back signals>
// === SUBTASK 3: Submodule Instantiations ===
// Register File
e203_exu_regfile u_e203_exu_regfile (
// <register file connections>
);
// Decode
e203_exu_decode u_e203_exu_decode (
// <decode connections>
);
// Dispatch
e203_exu_disp u_e203_exu_disp (
// <dispatch connections>
);
// OITF
e203_exu_oitf u_e203_exu_oitf (
// <OITF connections>
);
// ALU
e203_exu_alu u_e203_exu_alu (
// <ALU connections>
);
// Commit
e203_exu_commit u_e203_exu_commit (
// <commit connections>
);
// CSR
e203_exu_csr u_e203_exu_csr (
// <CSR connections>
);
// Long-pipeline Write-back
e203_exu_longpwbck u_e203_exu_longpwbck (
// <long-pipeline write-back connections>
);
// Write-back
e203_exu_wbck u_e203_exu_wbck (
// <write-back connections>
);
// === SUBTASK 4: Conditional Logic and Signal Assignments ===
// <exu_active signal logic>
// <dependency information logic>
// <optional NICE interface logic>
// <optional CSR_NICE interface logic>
endmodule
```

**Plan**

"subproblems": [
{ "id": "module_header_and_io", "description": "**Pseudocode Block**: This subtask implements the
[SUBTASK 1: Module Header and IO Port Definitions] block in the provided pseudocode. **Target
Implementation**: Define the complete module header with all interface signals including basic inter-
face, debug control interface, IFU IR stage interface, flush interface, LSU write-back interface, AGU
ICB interface, and optional CSR_NICE and NICE interfaces according to the specifications.", "re-
quired_section_indexes": [ "section_0", "section_1", "section_2", "section_3", "section_4", "section_5",
"section_6", "section_7", "section_8" ] },
{ "id": "internal_signals", "description": "**Pseudocode Block**: This subtask implements the
[SUBTASK 2: Internal Signal Declarations] block in the provided pseudocode. **Target Imple-
mentation**: Define all internal signals needed for connecting the submodules, including regis-
ter file interface signals, decode stage signals, dispatch stage signals, OITF signals, ALU signals,
commit stage signals, CSR signals, long-pipeline write-back signals, and write-back signals.", "re-
quired_section_indexes": [ "section_9", "section_10", "section_11", "section_12", "section_13", "sec-
tion_14", "section_15", "section_16", "section_17" ] },
...... ]

**Code Fragment**

```
// === SUBTASK 4: Conditional Logic and Signal Assignments ===
// exu_active signal logic
// Active when OITF is not empty, valid instruction entering, or exception active
assign exu_active = (~ oitf_empty) | i_valid | excp_active;
// dependency information logic for IFU
// dec2ifu_rden: Indicates whether current instruction writes to an integer register
assign dec2ifu_rden = dec_rdwen & (~dec_rdwen_fp);
// dec2ifu_rs1en: Indicates whether current instruction reads rs1 integer register
assign dec2ifu_rs1en = dec_rs1en & (~dec_rs1en_fp);
// dec2ifu_rdidx: Destination register index
assign dec2ifu_rdidx = dec_rdidx;
// rf2ifu_rs1: Value of rs1 register
assign rf2ifu_rs1 = read_src1_dat;
`ifdef E203_HAS_NICE
// optional NICE interface logic
assign nice_cmt_off_ilgl_o = nice_cmt_off_ilgl_i;
assign nice_xs_off = nice_xs_off_i;
`endif
`ifdef E203_HAS_CSR_NICE
// optional CSR_NICE interface logic
assign csr_nice_csr_valid = csr_nice_csr_valid_i;
assign csr_nice_csr_ready = csr_nice_csr_ready_i;
assign csr_nice_csr_addr = csr_nice_csr_addr_i;
assign csr_nice_csr_wr = csr_nice_csr_wr_i;
assign csr_nice_csr_wdata = csr_nice_csr_wdata_i;
assign csr_nice_csr_rdata = csr_nice_csr_rdata_i;
`endif
```

**Debug Action**

"fix_operations": [ { "operation": "delete_block", "start_line": 838, "end_line": 843 }, { "oper-
ation": "add_block", "line": 838, "content": [ " // Flush interface logic - handle different flush
conditions", " // Flush can be triggered by exceptions, interrupts, or control flow instructions", "
assign pipe_flush_req = commit_trap | commit_mret;", " ", " // For normal exceptions and inter-
rupts, use the trap vector", " // For returns from exceptions (mret), use the saved EPC", " assign
pipe_flush_add_op1 = commit_mret ? csr_epc_r : csr_mtvec_r;", " assign pipe_flush_add_op2 = 32'b0;",
"`ifdef E203_TIMING_BOOST", " assign pipe_flush_pc = commit_mret ? csr_epc_r : csr_mtvec_r;",
"`endif" ] }, { "operation": "delete_block", "start_line": 817, "end_line": 819 }, { "operation":
"add_block", "line": 817, "content": [ " // exu_active signal logic", " // Active when OITF is not
empty, valid instruction entering, or exception active", " // Also active during flush operations", " as-
sign exu_active = (~oitf_empty) | i_valid | excp_active | pipe_flush_req;" ] } ],

## C  FAILURE CASES

Although LocalV achieves significant performance gains over the baseline, it still faces persistent challenges in RealBench. These failure cases can be broadly categorized into three types: complex logic, syntactic errors, and excessive signals. In this section, we present examples of these three types of failure. Among these instances, the aes_cipher_top module failed to realize the complex encryption logic required. The e203_itcm_ctrl module encountered errors due to the improper application of macro expressions to define bit widths. In the case of e203_exu_alu_csrctrl, the failure was caused by incorrect use of comma triggered by an ifdef directive. Lastly, the e203_cpu_top module failed because the model confused signal directionality because of the excessive volume of signals.

---

**Failure Case: complex logic**

```
module aes_cipher_top(
......
);
......
// Instantiate 16 S-boxes for SubBytes transformation
aes_sbox u_sbox_00 (.a(sa00_r), .b(sa00_out));
......
// Instantiate key expansion module
aes_key_expand_128 u_key_expand (
......
);
......
// Combinational logic - state machine and transformations
always @(*) begin
......
case (state_r)
IDLE: begin
done_next = 1'b0;
if (ld) begin
state_next = INIT_ROUND;
sa00_next = text_in[127:120]; sa10_next = text_in[119:112];
......
INIT_ROUND: begin
......
ROUND_OP: begin
......
......
endmodule
```

---

**Failure Case: syntactic errors**

```
'include "e203_defines.v"
module e203_itcm_ctrl (
......
);
......
assign sram_icb_cmd_wdata = sram_sel_ifu ? ('E203_ITCM_DATA_WIDTH-'E203_XLEN)'b0,
ifu2itcm_icb_cmd_wdata :  arbt_icb_cmd_wdata;
assign sram_icb_cmd_wmask = sram_sel_ifu ? ('E203_ITCM_WMSK_WIDTH-'E203_XLEN/8)'b0,
ifu2itcm_icb_cmd_wmask :  arbt_icb_cmd_wmask;
assign sram_icb_cmd_size = sram_sel_ifu ? 2'b10 : arbt_icb_cmd_size; // IFU always uses word access
// Connect response signals from SRAM controller
assign ifu2itcm_icb_rsp_valid = sram_sel_ifu   sram_icb_rsp_valid;
assign arbt_icb_rsp_valid = sram_sel_arbt   sram_icb_rsp_valid;
assign sram_icb_rsp_ready = (sram_sel_ifu   ifu2itcm_icb_rsp_ready) |
(sram_sel_arbt   arbt_icb_rsp_ready);
......
endmodule
```

---

**Failure Case: syntactic errors**

```
'include "e203_defines.v"
module e203_exu_alu_csrctrl (
......
// Clock and reset
input wire clk,
input wire rst_n,
// NICE interface signals
'ifdef E203_HAS_CSR_NICE
// NICE interface signals
......
output wire [31:0] nice_csr_wdata,
input wire [31:0] nice_csr_rdata
'endif
);
......
endmodule
```

---

**Failure Case: excessive signals**

```
module e203_cpu_top (
......
// PPI ICB interface
input wire ppi_icb_cmd_valid,
output wire ppi_icb_cmd_ready,
input wire ['E203_ADDR_SIZE-1:0] ppi_icb_cmd_addr,
input wire ppi_icb_cmd_read,
......
);
......
e203_cpu u_e203_cpu (
// Clock and reset connections
.clk (clk),
.rst_n (rst_n),
......
// PPI ICB interface connections
.ppi_icb_enable (ppi_icb_enable),
.ppi_icb_cmd_valid (ppi_icb_cmd_valid),
.ppi_icb_cmd_ready (ppi_icb_cmd_ready),
.ppi_icb_cmd_addr (ppi_icb_cmd_addr),
.ppi_icb_cmd_read (ppi_icb_cmd_read),
......
)
......
endmodule
```

## D  DETAILS AND ANALYSIS OF THE DEBUG STEP

### D.1  DETAILED DEBUGGING WORKFLOW

In this section, we provide a comprehensive description of our iterative debugging workflow. To ensure reproducibility and clarity regarding the interaction between agents, the detailed procedure is outlined in Algorithm 1.

The process begins after the Merge Agent generates a candidate verilog code. The workflow proceeds as follows:

1. **Simulation:** We first compile and simulate the candidate code using the testbench provided by RealBench. If the simulation passes, the code is output as the final result.

2. **Fault Localization (AST-based):** Instead of feeding raw error logs directly to the LLM, we employ a Pyverilog-based AST method to trace the error signal back to its driver. This allows us to extract precise driver signals and their corresponding waveform information.

3. **Retrieval Augmented Context:** The localized AST guidance, along with error logs and the current code, is passed to the **Retriever**. The agent then queries the document descriptions to retrieve relevant reference sections.

4. **Debug Generation:** The **Debug Agent** receives a composite prompt containing the waveform information, error logs, code context, and retrieved documents. It then generates a specific edit action to fix the identified fault.

5. **Iteration:** The edit action is applied to the Verilog code, and the cycle repeats until the testbench passes or the maximum iteration limit is reached.

---

**Algorithm 1** Iterative Debugging with AST Guidance

---

**Input:** Verilog code $C_M$ from Merge Agent, testbench $TB$, document section descriptions $D$, max iterations $T_{max}$
**Output:** Verilog code after debug loop
1: $C_{curr} \leftarrow C_M$
2: $t \leftarrow 0$
3: **while** $t < T_{max}$ **do**
4:      Waveform, Errors, Pass $\leftarrow$ RunSimulation($C_{curr}, TB$)
5:      **if** Pass is **True then**
6:          **return** $C_{curr}$              ▷ Design verified successfully
7:      **end if**
8:                                ▷ Fault Localization via AST
9:      WaveformInfo $\leftarrow$ TraceAST($C_{curr}$, Errors)
10:                             ▷ Retrieval Step
11:      Query $\leftarrow$ {WaveformInfo, Errors, $C_{curr}$, $D$}
12:      Docs $\leftarrow$ RetrieverAgent(Query)
13:                                ▷ Debug Step
14:      Prompt $\leftarrow$ {WaveformInfo, Errors, $C_{curr}$, Docs}
15:      Action $\leftarrow$ DebugAgent(Prompt)
16:      $C_{curr} \leftarrow$ ApplyEdit($C_{curr}$, Action)
17:      $t \leftarrow t + 1$
18: **end while**
19: **return** $C_{curr}$             ▷ Return best effort if budget exhausted

---

## D.2 COST-BENEFIT ANALYSIS

To clarify the trade-off between computational cost and performance improvement, we analyze the average Pass@1 accuracy as a function of debugging iterations. Figure 7 illustrates the improvement in Pass@1 rate alongside the token usage per iteration.

## E DESIGN QUALITY

To quantitatively evaluate the hardware quality of the generated designs, we conducted a comprehensive PPA (Power, Performance, and Area) analysis. We utilized Yosys for logic synthesis to obtain the area usage, and employed OpenSTA to report the critical path delay and total power consumption. The generated Verilog code by LocalV was benchmarked against the golden implementations sourced from the RealBench dataset. Table 5 presents the detailed PPA comparison for each module. Since e203_extend_csr is an empty module, its metrics are null.

It can be observed from the table that for many test cases, the PPA metrics of the code generated by LocalV and the golden code are identical. This occurs because, although LocalV produces a functionally different implementation from the golden code, both are synthesized into the exact same hardware structure by the synthesis tool. To illustrate this, we present the code for the e203_exu_disp module, a case where the PPA results are identical. We have highlighted the distinct substructure in both our implementation and the golden code. Figure 8 and Figure 9 shows that these structurally different code segments result in identical synthesized netlist structure.

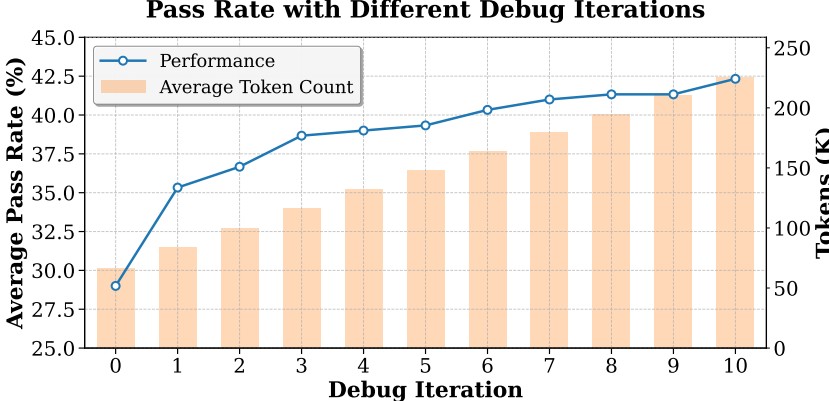

Figure 7: **Accuracy and Cost Trade-off.** The plot demonstrates the Pass@1 accuracy (left y-axis) and cumulative token usage (right y-axis) over 10 debug iterations. The most significant gain occurs in the first iteration.

Table 5: Design quality of LocalV vs. Golden RTL

| Design | Golden RTL | | | LocalV | | | Improvement | | |
|---|---|---|---|---|---|---|---|---|---|
| | Area ($\mu m^2$) | Delay (ns) | Power (mW) | Area ($\mu m^2$) | Delay (ns) | Power (mW) | Area | Delay | Power |
| aes_inv_sbox | 448.742 | 0.38 | 0.000654 | 448.742 | 0.38 | 0.000654 | 0.00% | 0.00% | 0.00% |
| aes_key_expand_128 | 3058.468 | 0.99 | 0.0193 | 2985.85 | 1.03 | 0.0177 | 2.37% | -4.04% | 8.29% |
| aes_sbox | 452.2 | 0.38 | 0.000665 | 623.77 | 0.42 | 0.000347 | -37.94% | -10.53% | 47.82% |
| e203_clk_ctrl | 31.92 | 0.52 | 5.22E-05 | 31.92 | 0.52 | 5.22E-05 | 0.00% | 0.00% | 0.00% |
| e203_clkgate | 2.128 | 0.04 | 1.18E-05 | 2.128 | 0.04 | 1.18E-05 | 0.00% | 0.00% | 0.00% |
| e203_dtcm_ctrl | 752.78 | 0.78 | 0.00125 | 758.366 | 0.78 | 0.00124 | -0.74% | 0.00% | 0.80% |
| e203_dtcm_ram | 4299995.07 | 0.14 | 99.3 | 4299995.07 | 0.14 | 99.3 | 0.00% | 0.00% | 0.00% |
| e203_exu_alu_bjp | 105.336 | 0.05 | 0.000529 | 105.336 | 0.05 | 0.000529 | 0.00% | 0.00% | 0.00% |
| e203_exu_alu_csrctrl | 138.054 | 0.31 | 0.00019 | 138.054 | 0.31 | 0.00019 | 0.00% | 0.00% | 0.00% |
| e203_exu_alu_rglr | 121.828 | 0.07 | 0.000435 | 121.828 | 0.07 | 0.000435 | 0.00% | 0.00% | 0.00% |
| e203_exu_decode | 576.688 | 0.79 | 0.000201 | 321.328 | 0.54 | 0.000132 | 44.28% | 31.65% | 34.33% |
| e203_exu_disp | 99.218 | 0.16 | 0.000158 | 99.218 | 0.16 | 0.000158 | 0.00% | 0.00% | 0.00% |
| e203_exu_nice | 156.674 | 0.37 | 0.000318 | 153.748 | 0.36 | 0.000603 | 1.87% | 2.70% | -89.62% |
| e203_exu_oitf | 743.736 | 0.45 | 0.00506 | 752.78 | 0.56 | 0.00414 | -1.22% | -24.44% | 18.18% |
| e203_exu_regfile | 8448.692 | 0.14 | 0.0939 | 8440.446 | 0.14 | 0.0939 | 0.10% | 0.00% | 0.00% |
| e203_ifu | 3106.082 | 4.58 | 0.0016 | 3106.082 | 3.83 | 0.00153 | 0.00% | 16.38% | 4.38% |
| e203_ifu_minidec | 576.688 | 0.79 | 0.000196 | 576.688 | 0.79 | 0.000196 | 0.00% | 0.00% | 0.00% |
| e203_irq_sync | 42.56 | 0.1 | 0.000371 | 42.56 | 0.1 | 0.000371 | 0.00% | 0.00% | 0.00% |
| e203_itcm_ram | 4285999.746 | 0.14 | 141 | 4285999.746 | 0.14 | 141 | 0.00% | 0.00% | 0.00% |
| e203_reset_ctrl | 12.502 | 0.1 | 0.000147 | 12.502 | 0.1 | 0.000147 | 0.00% | 0.00% | 0.00% |
| e203_srams | 8585994.816 | 0.14 | 238 | 8585995.348 | 0.14 | 208 | 0.00% | 0.00% | 12.61% |
| sd_clock_divider | 93.1 | 0.47 | 0.00061 | 93.1 | 0.47 | 0.00061 | 0.00% | 0.00% | 0.00% |
| sd_crc_16 | 128.212 | 0.23 | 0.00139 | 128.212 | 0.23 | 0.00139 | 0.00% | 0.00% | 0.00% |
| sd_crc_7 | 57.19 | 0.22 | 0.000646 | 57.19 | 0.22 | 0.000646 | 0.00% | 0.00% | 0.00% |
| sd_fifo_tx_filler | 2588.712 | 0.16 | 0.0467 | 2860.298 | 0.12 | 0.0639 | -10.49% | 25.00% | -36.83% |
| sd_tx_fifo | 2150.876 | 1.5 | 0.00554 | 1971.858 | 0.88 | 0.0072 | 8.32% | 41.33% | -29.96% |
| Average | 661380.0776 | 0.538461538 | 18.403074 | 661377.7757 | 0.481538462 | 17.24984931 | 0.25% | 3.00% | -1.15% |

---

Golden Case: e203_exu_disp

```
'include "e203_defines.v"
module e203_exu_disp(
  input wfi_halt_exu_req,
  output wfi_halt_exu_ack,
  input oitf_empty,
  input amo_wait,
  input disp_i_valid,
  output disp_i_ready,
  input disp_i_rs1x0,
  input disp_i_rs2x0,
  input disp_i_rs1en,
  input disp_i_rs2en,
  input ['E203_RFIDX_WIDTH-1:0] disp_i_rs1idx,
  input ['E203_RFIDX_WIDTH-1:0] disp_i_rs2idx,
  input ['E203_XLEN-1:0] disp_i_rs1,
  input ['E203_XLEN-1:0] disp_i_rs2,
  input disp_i_rdwen,
```

```
1026   Golden Case: e203_exu_disp
1027     input ['E203_RFIDX_WIDTH-1:0] disp_i_rdidx,
1028     input ['E203_DECINFO_WIDTH-1:0] disp_i_info,
1029     input ['E203_XLEN-1:0] disp_i_imm,
1030     input ['E203_PC_SIZE-1:0] disp_i_pc,
1031     input disp_i_misalgn,
1032     input disp_i_buserr ,
1033     input disp_i_ilegl ,
1034     output disp_o_alu_valid,
1035     input disp_o_alu_ready,
1036     input disp_o_alu_longpipe,
1037     output ['E203_XLEN-1:0] disp_o_alu_rs1,
1038     output ['E203_XLEN-1:0] disp_o_alu_rs2,
1039     output disp_o_alu_rdwen,
1040     output ['E203_RFIDX_WIDTH-1:0] disp_o_alu_rdidx,
1041     output ['E203_DECINFO_WIDTH-1:0] disp_o_alu_info,
1042     output ['E203_XLEN-1:0] disp_o_alu_imm,
1043     output ['E203_PC_SIZE-1:0] disp_o_alu_pc,
1044     output ['E203_ITAG_WIDTH-1:0] disp_o_alu_itag,
1045     output disp_o_alu_misalgn,
1046     output disp_o_alu_buserr ,
1047     output disp_o_alu_ilegl ,
1048     input oitfrd_match_disprs1,
1049     input oitfrd_match_disprs2,
1050     input oitfrd_match_disprs3,
1051     input oitfrd_match_disprd,
1052     input ['E203_ITAG_WIDTH-1:0] disp_oitf_ptr ,
1053     output disp_oitf_ena,
1054     input disp_oitf_ready,
1055     output disp_oitf_rs1fpu,
1056     output disp_oitf_rs2fpu,
1057     output disp_oitf_rs3fpu,
1058     output disp_oitf_rdfpu ,
1059     output disp_oitf_rs1en ,
1060     output disp_oitf_rs2en ,
1061     output disp_oitf_rs3en ,
1062     output disp_oitf_rdwen ,
1063     output ['E203_RFIDX_WIDTH-1:0] disp_oitf_rs1idx,
       output ['E203_RFIDX_WIDTH-1:0] disp_oitf_rs2idx,
       output ['E203_RFIDX_WIDTH-1:0] disp_oitf_rs3idx,
       output ['E203_RFIDX_WIDTH-1:0] disp_oitf_rdidx ,
       output ['E203_PC_SIZE-1:0] disp_oitf_pc ,
       input clk,
       input rst_n
       );
     wire ['E203_DECINFO_GRP_WIDTH-1:0] disp_i_info_grp = disp_i_info ['E203_DECINFO_GRP];
     wire disp_csr = (disp_i_info_grp == 'E203_DECINFO_GRP_CSR);
     wire disp_alu_longp_prdt = (disp_i_info_grp == 'E203_DECINFO_GRP_AGU)
                         ;
     wire disp_alu_longp_real = disp_o_alu_longpipe;
     wire disp_fence_fencei = (disp_i_info_grp == 'E203_DECINFO_GRP_BJP) &
                         ( disp_i_info ['E203_DECINFO_BJP_FENCE] | disp_i_info
['E203_DECINFO_BJP_FENCEI]);
     wire disp_i_valid_pos;
     wire disp_i_ready_pos = disp_o_alu_ready;
     assign disp_o_alu_valid = disp_i_valid_pos;
     wire raw_dep = ((oitfrd_match_disprs1) |
                 (oitfrd_match_disprs2) |
                 (oitfrd_match_disprs3));
     wire waw_dep = (oitfrd_match_disprd);
     wire dep = raw_dep | waw_dep;
     assign wfi_halt_exu_ack = oitf_empty & ( amo_wait);
```

```
Golden Case: e203_exu_disp

wire disp_condition =
            (disp_csr ? oitf_empty : 1'b1)
          & (disp_fence_fencei ? oitf_empty : 1'b1)
          & ( wfi_halt_exu_req)
          & ( dep)
          & (disp_alu_longp_prdt ? disp_oitf_ready : 1'b1);
assign disp_i_valid_pos = disp_condition & disp_i_valid;
assign disp_i_ready = disp_condition & disp_i_ready_pos;
wire ['E203_XLEN-1:0] disp_i_rs1_msked = disp_i_rs1 & {'E203_XLEN{ disp_i_rs1x0}};
wire ['E203_XLEN-1:0] disp_i_rs2_msked = disp_i_rs2 & {'E203_XLEN{ disp_i_rs2x0}};
assign disp_o_alu_rs1 = disp_i_rs1_msked;
assign disp_o_alu_rs2 = disp_i_rs2_msked;
assign disp_o_alu_rdwen = disp_i_rdwen;
assign disp_o_alu_rdidx = disp_i_rdidx;
assign disp_o_alu_info = disp_i_info;
assign disp_oitf_ena = disp_o_alu_valid & disp_o_alu_ready & disp_alu_longp_real;
assign disp_o_alu_imm = disp_i_imm;
assign disp_o_alu_pc = disp_i_pc;
assign disp_o_alu_itag = disp_oitf_ptr;
assign disp_o_alu_misalgn= disp_i_misalgn;
assign disp_o_alu_buserr = disp_i_buserr ;
assign disp_o_alu_ilegl = disp_i_ilegl ;
'ifndef E203_HAS_FPU
wire disp_i_fpu = 1'b0;
wire disp_i_fpu_rs1en = 1'b0;
wire disp_i_fpu_rs2en = 1'b0;
wire disp_i_fpu_rs3en = 1'b0;
wire disp_i_fpu_rdwen = 1'b0;
wire ['E203_RFIDX_WIDTH-1:0] disp_i_fpu_rs1idx = 'E203_RFIDX_WIDTH'b0;
wire ['E203_RFIDX_WIDTH-1:0] disp_i_fpu_rs2idx = 'E203_RFIDX_WIDTH'b0;
wire ['E203_RFIDX_WIDTH-1:0] disp_i_fpu_rs3idx = 'E203_RFIDX_WIDTH'b0;
wire ['E203_RFIDX_WIDTH-1:0] disp_i_fpu_rdidx = 'E203_RFIDX_WIDTH'b0;
wire disp_i_fpu_rs1fpu = 1'b0;
wire disp_i_fpu_rs2fpu = 1'b0;
wire disp_i_fpu_rs3fpu = 1'b0;
wire disp_i_fpu_rdfpu = 1'b0;
'endif
assign disp_oitf_rs1fpu = disp_i_fpu ? (disp_i_fpu_rs1en & disp_i_fpu_rs1fpu) : 1'b0;
assign disp_oitf_rs2fpu = disp_i_fpu ? (disp_i_fpu_rs2en & disp_i_fpu_rs2fpu) : 1'b0;
assign disp_oitf_rs3fpu = disp_i_fpu ? (disp_i_fpu_rs3en & disp_i_fpu_rs3fpu) : 1'b0;
assign disp_oitf_rdfpu = disp_i_fpu ? (disp_i_fpu_rdwen & disp_i_fpu_rdfpu ) : 1'b0;
assign disp_oitf_rs1en = disp_i_fpu ? disp_i_fpu_rs1en : disp_i_rs1en;
assign disp_oitf_rs2en = disp_i_fpu ? disp_i_fpu_rs2en : disp_i_rs2en;
assign disp_oitf_rs3en = disp_i_fpu ? disp_i_fpu_rs3en : 1'b0;
assign disp_oitf_rdwen = disp_i_fpu ? disp_i_fpu_rdwen : disp_i_rdwen;
assign disp_oitf_rs1idx = disp_i_fpu ? disp_i_fpu_rs1idx : disp_i_rs1idx;
assign disp_oitf_rs2idx = disp_i_fpu ? disp_i_fpu_rs2idx : disp_i_rs2idx;
assign disp_oitf_rs3idx = disp_i_fpu ? disp_i_fpu_rs3idx : 'E203_RFIDX_WIDTH'b0;
assign disp_oitf_rdidx = disp_i_fpu ? disp_i_fpu_rdidx : disp_i_rdidx;
assign disp_oitf_pc = disp_i_pc;
endmodule
```

```
LocalV Case: e203_exu_disp
```

```verilog
'include "e203_defines.v"
module e203_exu_disp (
    input clk,
    input rst_n,
    input wfi_halt_exu_req,
    output wfi_halt_exu_ack,
    input oitf_empty,
    input amo_wait,
    input ['E203_ITAG_WIDTH-1:0] disp_oitf_ptr,
    output disp_oitf_ena,
    input disp_oitf_ready,
    output disp_oitf_rs1fpu,
    output disp_oitf_rs2fpu,
    output disp_oitf_rs3fpu,
    output disp_oitf_rdfpu,
    output disp_oitf_rs1en,
    output disp_oitf_rs2en,
    output disp_oitf_rs3en,
    output disp_oitf_rdwen,
    output ['E203_RFIDX_WIDTH-1:0] disp_oitf_rs1idx,
    output ['E203_RFIDX_WIDTH-1:0] disp_oitf_rs2idx,
    output ['E203_RFIDX_WIDTH-1:0] disp_oitf_rs3idx,
    output ['E203_RFIDX_WIDTH-1:0] disp_oitf_rdidx,
    output ['E203_PC_SIZE-1:0] disp_oitf_pc,
    input disp_i_valid,
    output disp_i_ready,
    input disp_i_rs1x0,
    input disp_i_rs2x0,
    input disp_i_rs1en,
    input disp_i_rs2en,
    input ['E203_RFIDX_WIDTH-1:0] disp_i_rs1idx,
    input ['E203_RFIDX_WIDTH-1:0] disp_i_rs2idx,
    input ['E203_XLEN-1:0] disp_i_rs1,
    input ['E203_XLEN-1:0] disp_i_rs2,
    input disp_i_rdwen,
    input ['E203_RFIDX_WIDTH-1:0] disp_i_rdidx,
    input ['E203_DECINFO_WIDTH-1:0] disp_i_info,
    input ['E203_XLEN-1:0] disp_i_imm,
    input ['E203_PC_SIZE-1:0] disp_i_pc,
    input disp_i_misalgn,
    input disp_i_buserr,
    input disp_i_ilegl,
    output disp_o_alu_valid,
    input disp_o_alu_ready,
    input disp_o_alu_longpipe,
    output ['E203_XLEN-1:0] disp_o_alu_rs1,
    output ['E203_XLEN-1:0] disp_o_alu_rs2,
    output disp_o_alu_rdwen,
    output ['E203_RFIDX_WIDTH-1:0] disp_o_alu_rdidx,
    output ['E203_DECINFO_WIDTH-1:0] disp_o_alu_info,
    output ['E203_XLEN-1:0] disp_o_alu_imm,
    output ['E203_PC_SIZE-1:0] disp_o_alu_pc,
    output ['E203_ITAG_WIDTH-1:0] disp_o_alu_itag,
    output disp_o_alu_misalgn,
    output disp_o_alu_buserr,
    output disp_o_alu_ilegl,
    input oitfrd_match_disprs1,
    input oitfrd_match_disprs2,
    input oitfrd_match_disprs3,
    input oitfrd_match_disprd
);
```

```
LocalV Case: e203_exu_disp

    wire disp_csr = (disp_i_info['E203_DECINFO_GRP] == 'E203_DECINFO_GRP_CSR);
    wire disp_agu = (disp_i_info['E203_DECINFO_GRP] == 'E203_DECINFO_GRP_AGU);
    wire disp_bjp = (disp_i_info['E203_DECINFO_GRP] == 'E203_DECINFO_GRP_BJP);
    wire disp_fence = disp_bjp & disp_i_info['E203_DECINFO_BJP_FENCE];
    wire disp_fencei = disp_bjp & disp_i_info['E203_DECINFO_BJP_FENCEI];
    wire need_wait_oitf_empty = disp_csr | disp_fence | disp_fencei;
    wire rs1_dep_oitf_idx = oitfrd_match_disprs1;
    wire rs2_dep_oitf_idx = oitfrd_match_disprs2;
    wire rs3_dep_oitf_idx = oitfrd_match_disprs3;
    wire disp_raw_dep = rs1_dep_oitf_idx | rs2_dep_oitf_idx | rs3_dep_oitf_idx;
    wire disp_waw_dep = oitfrd_match_disprd;
    wire ['E203_XLEN-1:0] disp_i_rs1_msked = disp_i_rs1x0 ? {'E203_XLEN{1'b0}} : disp_i_rs1;
    wire ['E203_XLEN-1:0] disp_i_rs2_msked = disp_i_rs2x0 ? {'E203_XLEN{1'b0}} : disp_i_rs2;
    wire oitf_empty_condition = ( need_wait_oitf_empty) | (need_wait_oitf_empty & oitf_empty);
    wire no_dep_condition = ( disp_raw_dep) & ( disp_waw_dep);
    wire wfi_halt_condition =  wfi_halt_exu_req;
    wire agu_oitf_ready = ( disp_agu) | (disp_agu & disp_oitf_ready);
    wire disp_condition = oitf_empty_condition & wfi_halt_condition & no_dep_condition &
 agu_oitf_ready;
    assign disp_i_ready = disp_condition & disp_o_alu_ready;
    assign disp_oitf_ena = disp_o_alu_longpipe & disp_o_alu_valid & disp_o_alu_ready;
    assign disp_o_alu_valid = disp_condition & disp_i_valid;
    assign disp_o_alu_rs1 = disp_i_rs1_msked;
    assign disp_o_alu_rs2 = disp_i_rs2_msked;
    assign disp_o_alu_rdwen = disp_i_rdwen;
    assign disp_o_alu_rdidx = disp_i_rdidx;
    assign disp_o_alu_info = disp_i_info;
    assign disp_o_alu_imm = disp_i_imm;
    assign disp_o_alu_pc = disp_i_pc;
    assign disp_o_alu_itag = disp_oitf_ptr;
    assign disp_o_alu_misalgn = disp_i_misalgn;
    assign disp_o_alu_buserr = disp_i_buserr;
    assign disp_o_alu_ilegl = disp_i_ilegl;
    assign disp_oitf_rs1en = disp_i_rs1en;
    assign disp_oitf_rs2en = disp_i_rs2en;
    assign disp_oitf_rdwen = disp_i_rdwen;
    assign disp_oitf_rs1idx = disp_i_rs1idx;
    assign disp_oitf_rs2idx = disp_i_rs2idx;
    assign disp_oitf_rdidx = disp_i_rdidx;
    assign disp_oitf_pc = disp_i_pc;
    assign disp_oitf_rs3en = 1'b0;
    assign disp_oitf_rs3idx = {'E203_RFIDX_WIDTH{1'b0}};
'ifdef E203_HAS_FPU
    assign disp_oitf_rs1fpu = 1'b0;
    assign disp_oitf_rs2fpu = 1'b0;
    assign disp_oitf_rs3fpu = 1'b0;
    assign disp_oitf_rdfpu = 1'b0;
'else
    assign disp_oitf_rs1fpu = 1'b0;
    assign disp_oitf_rs2fpu = 1'b0;
    assign disp_oitf_rs3fpu = 1'b0;
    assign disp_oitf_rdfpu = 1'b0;
'endif
    assign wfi_halt_exu_ack = oitf_empty & ( amo_wait);
endmodule
```

## F    INFORMATION LOCALITY CASE

In this section, we present e203_srams as an instance of high information locality, while contrasting it with Parse Lisp Expression, which exhibits low information locality.

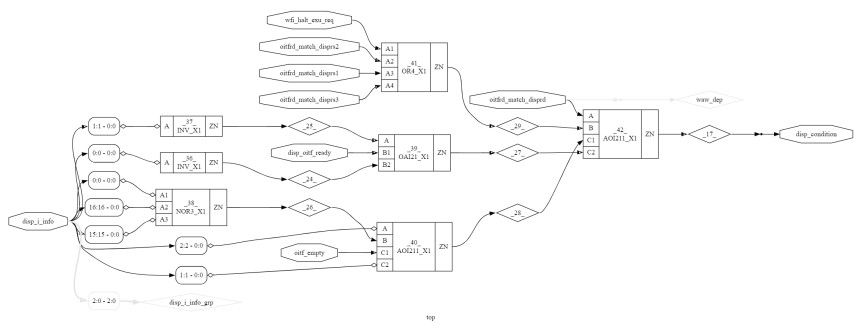

Figure 8: The netlist of the substructure of golden e203_exu_disp module after synthesis

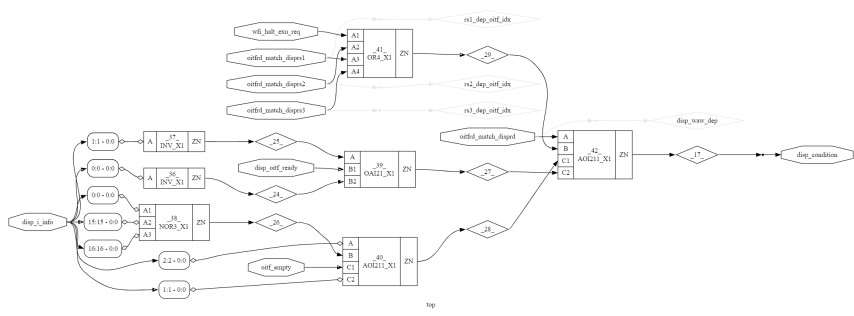

Figure 9: The netlist of the substructure of LocalV's e203_exu_disp module after synthesis

The e203_srams specification from RealBench exhibits clear modularity. The functional description and interface definitions of each task (ITCM RAM and DTCM RAM) are grouped tightly together in dedicated sections. This allows sub-tasks to be implemented using only partial, relevant documentation without interference from other sub-modules.

Conversely, the Parse Lisp Expression task demonstrates weaker information locality. The description is broad, making it hard to pinpoint specific paragraphs that correspond to the code's abstract algorithms. For example, the stack structure required for implementation has no corresponding section in the document; instead, it must be abstracted from the overall problem description. As a result, the full document is usually required to understand and design the complete algorithm.

---

**Good Case: e203_srams document**

# e203_srams Design Documentation
## 1. Introduction
The e203_srams module is the memory management module of the E203 processor. It is mainly used for integrating and managing the Instruction Tightly Coupled Memory (ITCM) and Data Tightly Coupled Memory (DTCM). This module flexibly controls the instantiation of ITCM and DTCM through macro definitions 'E203_HAS_ITCM' and 'E203_HAS_DTCM'.
## 2. Module Block Diagram

## 3. Interface Definition
### General Interface
| Signal Name | Direction | Bit Width | Description |
|———|———|———|———|
| test_mode | Input | 1 | Unused and unassigned |

---

**Good Case: e203_srams document**

### ITCM RAM Interface Signals exist only if the 'E203_HAS_ITCM' is defined

| Signal Name | Direction | Bit Width | Description |
|——|——|——|——|
| itcm_ram_sd | Input | 1 | ITCM power off enable signal |
| itcm_ram_ds | Input | 1 | ITCM deep sleep mode enable |
| itcm_ram_ls | Input | 1 | ITCM light sleep mode enable |
| itcm_ram_cs | Input | 1 | ITCM chip select signal |
| itcm_ram_we | Input | 1 | ITCM write enable signal |
| itcm_ram_addr | Input | E203_ITCM_RAM_AW | ITCM address |
| itcm_ram_wem | Input | E203_ITCM_RAM_MW | ITCM write mask |
| itcm_ram_din | Input | E203_ITCM_RAM_DW | ITCM write data |
| itcm_ram_dout | Output | E203_ITCM_RAM_DW | ITCM read data |
| clk_itcm_ram | Input | 1 | ITCM clock signal |
| rst_itcm | Input | 1 | ITCM reset signal |

### DTCM RAM Interface
Signals exist only if the 'E203_HAS_DTCM' is defined
(Similar to the ITCM interface, with the signal name prefix changed to dtcm).

| Signal Name | Direction | Bit Width | Description |
|——|——|——|——|
| dtcm_ram_sd | Input | 1 | DTCM power off enable signal |
| dtcm_ram_ds | Input | 1 | DTCM deep sleep mode enable |
| dtcm_ram_ls | Input | 1 | DTCM light sleep mode enable |
| dtcm_ram_cs | Input | 1 | DTCM chip select signal |
| dtcm_ram_we | Input | 1 | DTCM write enable signal |
| dtcm_ram_addr | Input | E203_ITCM_RAM_AW | DTCM address |
| dtcm_ram_wem | Input | E203_ITCM_RAM_MW | DTCM write mask |
| dtcm_ram_din | Input | E203_ITCM_RAM_DW | DTCM write data |
| dtcm_ram_dout | Output | E203_ITCM_RAM_DW | DTCM read data |
| clk_dtcm_ram | Input | 1 | DTCM clock signal |
| rst_dtcm | Input | 1 | DTCM reset signal |

## 4. Submodule List

### ITCM RAM

#### Function

The e203_dtcm_ram module is a Data Tightly Coupled Memory (DTCM) RAM module for the E203 processor. The module is encapsulated based on a generic RAM module, primarily used for data storage and access. The module is controlled by the macro definition 'E203_HAS_DTCM'.

#### Interface

| Signal Name | Direction | Width | Description |
|——|——|——|——|
| sd | Input | 1 | Power domain shutdown enable signal for power management |
| ds | Input | 1 | Deep sleep mode enable, controlling complete power area shutdown |
| ls | Input | 1 | Light sleep mode enable, reducing power without full shutdown |
| cs | Input | 1 | Chip select signal, controlling RAM selection |
| we | Input | 1 | Write enable signal, controlling write operation |
| addr | Input | E203_ITCM_RAM_AW | Address input, specifying read/write location |
| wem | Input | E203_ITCM_RAM_MW | Write mask, controlling specific byte writing |
| din | Input | E203_ITCM_RAM_DW | Data input to be written |
| rst_n | Input | 1 | Asynchronous reset signal (active low) |
| clk | Input | 1 | System clock |
| dout | Output | E203_ITCM_RAM_DW | Data output, read data |

### DTCM RAM

#### Function

The e203_itcm_ram module is an Instruction Tightly Coupled Memory (ITCM) RAM module for the E203 processor. The module is encapsulated based on a generic RAM module, primarily used for instruction storage and access. The module is controlled by the macro definition 'E203_HAS_ITCM'.

#### Interface

| Signal Name | Direction | Width | Description |
|——|——|——|——|
| sd | Input | 1 | Power domain shutdown enable signal for power management |
| ds | Input | 1 | Deep sleep mode enable, controlling complete power area shutdown |
| ls | Input | 1 | Light sleep mode enable, reducing power without full shutdown |
| cs | Input | 1 | Chip select signal, controlling RAM selection |

---

**Good Case: e203_srams document**

| we | Input | 1 | Write enable signal, controlling write operation |
| addr | Input | E203_DTCM_RAM_AW | Address input, specifying read/write location |
| wem | Input | E203_DTCM_RAM_MW | Write mask, controlling specific byte writing |
| din | Input | E203_DTCM_RAM_DW | Data input to be written |
| rst_n | Input | 1 | Asynchronous reset signal (active low) |
| clk | Input | 1 | System clock |
| dout | Output | E203_DTCM_RAM_DW | Data output, read data |

## 5. Implementation Details
1. Memory management mechanism
- Supports independent configuration and control of ITCM and DTCM.
- Each memory module has an independent clock and reset domain.
2. Data flow control
- Adopts a preprocessed data output mechanism (dout_pre).
- Removes the data bypass function in test mode.
3. Submodule Instantiation Details
The submodule interface is connected to the corresponding interface of this module. For example, the 'sd' signal of 'e203_itcm_ram' is connected to the 'itcm_ram_sd' interface.
## 6. Limitations
1. Functional constraints
- The address must be within the valid range.

---

**Good Case: e203_srams code**

```
'include "e203_defines.v"
module e203_srams(
    ......
);
'ifdef E203_HAS_ITCM //
wire ['E203_ITCM_RAM_DW-1:0] itcm_ram_dout_pre;
e203_itcm_ram u_e203_itcm_ram (
    .sd (itcm_ram_sd),
    .ds (itcm_ram_ds),
    .ls (itcm_ram_ls),
    .cs (itcm_ram_cs ),
    .we (itcm_ram_we ),
    .addr (itcm_ram_addr ),
    .wem (itcm_ram_wem ),
    .din (itcm_ram_din ),
    .dout (itcm_ram_dout_pre ),
    .rst_n(rst_itcm ),
    .clk (clk_itcm_ram )
    );
    assign itcm_ram_dout = itcm_ram_dout_pre;
'endif//
'ifdef E203_HAS_DTCM //
wire ['E203_DTCM_RAM_DW-1:0] dtcm_ram_dout_pre;
e203_dtcm_ram u_e203_dtcm_ram (
    .sd (dtcm_ram_sd),
    .ds (dtcm_ram_ds),
    .ls (dtcm_ram_ls),
    .cs (dtcm_ram_cs ),
    .we (dtcm_ram_we ),
    .addr (dtcm_ram_addr ),
    .wem (dtcm_ram_wem ),
    .din (dtcm_ram_din ),
    .dout (dtcm_ram_dout_pre ),
    .rst_n(rst_dtcm ),
    .clk (clk_dtcm_ram )
    );
    assign dtcm_ram_dout = dtcm_ram_dout_pre;
'endif//
endmodule
```

---

**Bad Case: Parse Lisp Expression document**

You are given a string expression representing a Lisp-like expression to return the integer value of. The syntax for these expressions is given as follows.

An expression is either an integer, let expression, add expression, mult expression, or an assigned variable. Expressions always evaluate to a single integer. (An integer could be positive or negative.)

A let expression takes the form "(let v1 e1 v2 e2 ... vn en expr)", where let is always the string "let", then there are one or more pairs of alternating variables and expressions, meaning that the first variable v1 is assigned the value of the expression e1, the second variable v2 is assigned the value of the expression e2, and so on sequentially; and then the value of this let expression is the value of the expression expr.

An add expression takes the form "(add e1 e2)" where add is always the string "add", there are always two expressions e1, e2 and the result is the addition of the evaluation of e1 and the evaluation of e2.

A mult expression takes the form "(mult e1 e2)" where mult is always the string "mult", there are always two expressions e1, e2 and the result is the multiplication of the evaluation of e1 and the evaluation of e2.

For this question, we will use a smaller subset of variable names. A variable starts with a lowercase letter, then zero or more lowercase letters or digits. Additionally, for your convenience, the names "add", "let", and "mult" are protected and will never be used as variable names.

Finally, there is the concept of scope. When an expression of a variable name is evaluated, within the context of that evaluation, the innermost scope (in terms of parentheses) is checked first for the value of that variable, and then outer scopes are checked sequentially. It is guaranteed that every expression is legal. Please see the examples for more details on the scope.

Example 1:

Input: expression = "(let x 2 (mult x (let x 3 y 4 (add x y))))" Output: 14

Explanation: In the expression (add x y), when checking for the value of the variable x, we check from the innermost scope to the outermost in the context of the variable we are trying to evaluate. Since x = 3 is found first, the value of x is 3.

Example 2:

Input: expression = "(let x 3 x 2 x)"

Output: 2

Explanation: Assignment in let statements is processed sequentially.

Example 3:

Input: expression = "(let x 1 y 2 x (add x y) (add x y))"

Output: 5

Explanation: The first (add x y) evaluates as 3, and is assigned to x. The second (add x y) evaluates as 3+2 = 5.

Constraints:

1 <= expression.length <= 2000

There are no leading or trailing spaces in expression. All tokens are separated by a single space in expression. The answer and all intermediate calculations of that answer are guaranteed to fit in a 32-bit integer. The expression is guaranteed to be legal and evaluate to an integer.

---

```
Bad Case: Parse Lisp Expression code

def implicit_scope(func):
    def wrapper(*args):
        args[0].scope.append()
        ans = func(*args)
        args[0].scope.pop()
        return ans
    return wrapper
class Solution(object):
    def __init__(self):
        self.scope = []
    @implicit_scope
    def evaluate(self, expression):
        if not expression.startswith('('):
            if expression[0].isdigit() or expression[0] == '-':
                return int(expression)
            for local in reversed(self.scope):
                if expression in local: return local[expression]
        tokens = list(self.parse(expression[5 + (expression[1] == 'm'): -1]))
        if expression.startswith('(add'):
            return self.evaluate(tokens[0]) + self.evaluate(tokens[1])
        elif expression.startswith('(mult'):
            return self.evaluate(tokens[0]) * self.evaluate(tokens[1])
        else:
            for j in xrange(1, len(tokens), 2):
                self.scope[-1][tokens[j-1]] = self.evaluate(tokens[j])
            return self.evaluate(tokens[-1])
    def parse(self, expression):
        bal = 0
        buf = []
        for token in expression.split():
            bal += token.count('(') - token.count(')')
            buf.append(token)
            if bal == 0:
                yield " ".join(buf)
                buf = []
        if buf:
            yield " ".join(buf)
```

# G   LLM USAGE

Large language models (LLMs) were utilized to assist in the writing and polishing of this manuscript. Specifically, LLMs were employed to help refine language, improve readability, and enhance clarity across various sections of the paper. This included tasks such as rephrasing sentences, checking grammar, and improving the overall coherence and flow of the text.

