# OpenReview forum: "LocalV: Exploiting Information Locality for IP-level Verilog Generation"
_ICLR.cc/2026/Conference — Submitted to ICLR 2026_

### Official Review · Reviewer_QRyn · 2025-10-30

**Soundness:** 3
**Presentation:** 3
**Contribution:** 3
**Rating:** 6
**Confidence:** 4

**Summary:**

This paper introduces LocalV, a multi-agent framework designed for IP-level Verilog generation that leverages the inherent information locality of modular hardware design. Instead of tackling the long-document to long-code generation challenge directly, LocalV decomposes it into a series of manageable short-document to short-code subtasks, thereby enhancing scalability in both generation and debugging. The framework incorporates three core innovations: a hierarchical indexing mechanism, a fragment-oriented task decomposition scheme, and a locality-aware debugging loop. Experimental results on the RealBench benchmark demonstrate that LocalV significantly surpasses existing state-of-the-art LLMs and agent-based systems, underscoring its potential for automating IP-level RTL design.

**Strengths:**

This paper contributes to automated hardware design by addressing the practical challenge of IP-level Verilog generation. The proposed LocalV framework introduces a coherent and technically grounded approach that leverages information locality in modular hardware design to decompose long-document to long-code generation into smaller, locality-aware subtasks. Its design combines a hierarchical indexing mechanism, fragment-based task decomposition, and a locality-guided debugging process, offering a structured solution to scalability and correctness issues in IP-level RTL synthesis. The experimental evaluation on RealBench provides evidence of LocalV’s effectiveness, showing consistent improvements over existing LLMs and agent-based methods. The paper is clearly presented and logically organized, making its motivation, methodology, and results easy to follow. Overall, it offers a meaningful step toward scalable and reliable RTL generation for practical IP-level design tasks.

**Weaknesses:**

While the paper introduces a promising framework, its technical differentiation from prior agent-based systems is not entirely clear. The authors emphasize that LocalV avoids cascading ambiguity by maintaining a direct link between specifications and code, but it remains uncertain whether this alone constitutes a substantial methodological advancement over existing frameworks such as MAGE or Spec2RTL-Agent. A more detailed comparison, highlighting architectural distinctions, workflow advantages, or specific design decisions beyond the specification-to-code linkage, would help clarify the scope of novelty. Furthermore, the approach fundamentally relies on the information locality hypothesis, yet the quantitative results in Section 3.2 show a noticeable gap in normalized entropy between idealized and real-world cases, suggesting that the assumption may not consistently hold across complex or loosely structured specifications. This raises concerns about robustness and generalizability of the proposed method. Lastly, the paper lacks a reproducibility statement or code-release plan, which limits the ability of other researchers to validate or extend the work.

**Questions:**

1. In Section 3.2, the authors use Qwen3-Embedding-0.6B to compute semantic similarities between specification and code fragments when validating the locality assumption. Could the authors elaborate on the rationale behind this choice? Have other embedding models been considered to verify the robustness of this assumption? Given that Qwen3-Embedding-0.6B may have limited understanding of Verilog semantics, it might produce misleading similarity scores. There are domain-specific embedding models such as DeepRTL2[1] that are trained on RTL data. Have the authors explored or considered using such models to obtain more accurate and representative results?

    [1] Liu, Yi, et al. "DeepRTL2: A Versatile Model for RTL-Related Tasks.” *Findings of the Association for Computational Linguistics: ACL 2025*. 2025.

2. The paper mentions that the raw specification documents are split into *coherent paragraphs* before hierarchical indexing. Could the authors clarify how this segmentation is performed? Is it based purely on structural markers or on semantic coherence using an LLM?
3. During the planning and task decomposition stage, the retriever is responsible for fetching the most relevant document fragments for each subtask. Have the authors quantitatively evaluated the retriever’s accuracy? Since errors at this stage could propagate through the generation pipeline, an assessment of retrieval precision and recall would strengthen confidence in the overall system reliability.
4. Although LocalV outperforms existing baselines, the overall functional accuracy on RealBench remains modest. Have the authors conducted any failure analysis to identify common error patterns or limitations in the failed cases? Insights into where LocalV struggles would help clarify the boundaries of the proposed method’s effectiveness and inform directions for future improvement.

---

> ### Author Response · Authors · 2025-11-22
> **Response to Reviewer QRyn (1/2)**
>
> Thank you for your valuable feedback. We appreciate your recognition of our contributions, including addressing the practical challenge of IP-level Verilog generation, coherent and technically grounded method, and a meaningful step for practical IP-level design tasks. Your insights on the limitations, including further comparisons with previous work, analysis on the information locality hypothesis and retriever, and request for the plan of open-source, have guided our improvements. We have added comprehensive experiments, provided further analysis, and revised the paper accordingly. We will address your concerns as follows:
>
> > **W1. While the paper introduces a promising framework, its technical differentiation from prior agent-based systems is not entirely clear.**
>
> We thank the reviewer for the insightful comparison. The goal of Spec2RTL-Agent is to assist humans in completing the design, so its primary evaluation metric is the number of human interventions required. In contrast, our goal is to automatically design IP-level designs, making the two efforts parallel rather than comparable. As for VerilogCoder and MAGE, they also focus on automatic design, and we discuss the differences between our work and theirs in the following table:
>
> | Component | LocalV | VerilogCoder | MAGE |
> | - | - | - | - |
> | Plan Agent | $\checkmark$ | $\checkmark$ | $\times$ |
> | **Indexing Original Document During Planning** | $\checkmark$ | $\times$ | $\times$ |
> | Code Agent | $\checkmark$ | $\checkmark$ | $\checkmark$ |
> | Merge Agent | LLM | Rule | $\times$ |
> | Debug Agent | $\checkmark$ | $\checkmark$ | $\checkmark$ |
> | **Indexing Original Document During Debugging** | $\checkmark$ | $\times$ | $\times$ |
>
> From this table, we can clearly see that MAGE does not have the decomposing step compared to LocalV. Meanwhile, although VerilogCoder has a decompose step, LocalV introduces **original document indexing during both planning and debugging**, and **LLM-based code merging** to address the challenges of **IP-level Verilog generation**.
>
> For **indexing original document**, it is the core of LocalV to **utilize locality** to address the **excessive context length** problem in complex IP design. In our ablation study (Table 4 in the new paper), removing indexing in planning decreases the functional accuracy **from 45% to 35%**. In addition, we have conducted another ablation study to remove indexing in debugging (please refer to W5 by Reviewer aBM8 for details), and found that it decreases the functional accuracy **from 45% to 38.3%**. From these ablations, it is clear that locality is essential to solve long problems.
>
> For **LLM-based code merging**, it is more flexible by leaving the module merging to LLM compared to partitioning problems into abstract sub-modules with rigid interfaces in VerilogCoder. The code agent only needs to implement the core Verilog logic (e.g., wiring, independent blocks) without the need of VerilogCoder's Verilog verification assistant to check module syntax.
>
> ---
> > **W2. Furthermore, the approach fundamentally relies on the information locality hypothesis, yet the quantitative results in Section 3.2 show a noticeable gap in normalized entropy between idealized and real-world cases, suggesting that the assumption may not consistently hold across complex or loosely structured specifications.**
>
> We appreciate this question! Our planner agent and retriever are fault-tolerant: they can flexibly retrieve the top-ranked document sections based on the document, so the documents do not need to strictly follow the specification for the system to perform well. Our results on RealBench (improving over the strong Agent baseline by 23.4%) already demonstrate this.
>
> To further validate the method’s tolerance to imperfect document specification, we conducted an additional experiment. We deliberately damaged the document's information locality by randomly combining paragraphs (i.e., we made the document "flatten"). Specifically, we conduct experiments on RealBench by first setting ratios and then combining paragraphs to make the total number of paragraphs to that certain ratio. Then, we use the same workflow of LocalV to generate RTL based on these modified documents. The results are shown in the table below:
>
> | Ratio | Accuracy |
> | - | - |
> | 1x (ours) | 45.0% |
> | 0.8x | 43.3% |
> | 0.7x | 38.3% |
>
> The results show that even when the document is flattened to 0.8, our method still maintains good accuracy. This is because we allow some flexibility during document retrieval, enabling us to handle documents that are not well-structured to some extent. When the document is excessively flattened, such as down to 0.7, the performance drops somewhat, but it still remains 16.7% higher than the baseline MAGE.

---

> ### Author Response · Authors · 2025-11-22
> **Response to Reviewer QRyn (2/2)**
>
> > **W3. The paper lacks a reproducibility statement or code-release plan.**
>
> We appreciate this question! We plan to release the complete LocalV implementation on GitHub soon after the review period to facilitate community validation and future research.
>
> ---
> > **Q1. Have the authors explored or considered using such models to obtain more accurate and representative results?**
>
> We appreciate this question! We choose Qwen3-Embedding-0.6B because, in validating the information locality hypothesis, we care about relative magnitudes across different kinds of modules rather than absolute values. Therefore, the 0.6B model is sufficient for validating our hypothesis while also being computationally efficient. To further address your concern, we evaluated the results of BM25, the DeepRTL2 (Llama) model, and the Qwen3-Embedding-8B model. As shown in the table below, the different models exhibit consistent behavior on these circuits: the locality of the 10 Combined Modules is higher than that of the individual circuit modules, which in turn is higher than that of software code. We have updated these results in the paper to further support the information locality hypothesis.
>
> | Module | BM25 | DeepRTL2 | Qwen-0.6B | Qwen-8B |
> | :--- | :--- | :--- | :--- | :--- |
> | E203 CPU Top Module | 0.4036 | 0.9504 | 0.8680 | 0.8873 |
> | Parse Lisp Expression | 0.5412 | 0.9699 | 0.9126 | 0.9453 |
> | 10 Combined Modules | 0.0669 | 0.9255 | 0.8206 | 0.8167 |
>
> ---
> > **Q2. The paper mentions that the raw specification documents are split into coherent paragraphs before hierarchical indexing. Could the authors clarify how this segmentation is performed? Is it based purely on structural markers or on semantic coherence using an LLM?**
>
> We appreciate this question! In our current implementation, the partitioning mechanism is straightforward yet effective, leveraging the inherent modularity commonly found in IP-level specification documents. We simply partition the document based on its semantic section markers (e.g., "##" in Markdown). Each resulting section is treated as a distinct chunk of information.
>
> ---
> > **Q3. Since errors at this stage could propagate through the generation pipeline, an assessment of retrieval precision and recall would strengthen confidence in the overall system reliability.**
>
> We appreciate this question! To measure the retriever’s accuracy, we manually annotated the golden document fragments that should be retrieved for the AES IP in RealBench and used them as a test set to evaluate retrieval quality. The results in the table below show that both LocalV (Claude) and LocalV (DeepSeek-V3) achieve high retrieval accuracy. This demonstrates that the indexing step is not a performance bottleneck, as our design supports consistent retrieval quality across models with different capabilities.
>
> | |Precision|Recall|
> |-|-|-|
> |LocalV (DeepSeek-V3)|0.93|0.93|
> |LocalV (Claude)|0.92|0.95|
>
> ---
> > Q4. Have the authors conducted any failure analysis to identify common error patterns or limitations in the failed cases?
>
> We appreciate this constructive suggestion. In response, we have conducted a qualitative analysis of failure cases and included a detailed discussion in **Appendix C** of the revised manuscript.
> Our investigation reveals that failures generally stem from three scenarios. First, regarding functional correctness, the model faces difficulties with complex logic, it may fail to generate correct encryption sequences for modules like aes_cipher_top. Second, syntactic errors may occur when handling bit-width definitions that rely on complex macro expressions. Finally, we observed limitations in handling large-scale signals, where an excessive number of signals can overwhelm the model, leading to incorrect signal assignments.
>
> ---
> Thank you again for your valuable feedback. We hope the rebuttal is adequate in addressing your concerns, and if so, please update your review in accordance. If any questions remain, we welcome further discussion.

---

### Official Review · Reviewer_LYBq · 2025-10-31

**Soundness:** 3
**Presentation:** 3
**Contribution:** 3
**Rating:** 6
**Confidence:** 5

**Summary:**

This paper proposes LocalV, a multi-agent framework for generating IP-level Verilog code from natural language specifications. The key idea is to leverage the information locality inherent in modular hardware design -- where each code fragment primarily depends on a small, corresponding portion of the specification. LocalV decomposes long-document to long-code generation into manageable sub-tasks, integrating hierarchical document indexing, structured planning, localized generation, merging with interface consistency, and locality-aware debugging.

**Strengths:**

1. The information locality is intuitive. By grounding the workflow in the modular structure of hardware specifications, this paper establishes a conceptual foundation for decomposing long, complex design documents into smaller, tractable tasks.

**Weaknesses:**

1. The granularity of document partitioning and task decomposition is unclear. Since different hardware modules can span multiple paragraphs or share interfaces, it is unclear how partition boundaries are chosen or adapted to maintain coherence across related fragments.
2. The merging and debugging stages appear to rely primarily on observations rather than formal verification. Without stronger guarantees of merging correctness or synthesized design quality (e.g., timing, area), it is difficult to assess how close LocalV comes to generating deployable industrial RTL.

**Questions:**

1. Does partition granularity impact performance? How does LocalV ensure coherence when multiple paragraphs describe interdependent modules?
2. Are the extracted paragraphs validated as meaningful semantic units, or could irrelevant segments mislead sub-task generation?
3. How is correctness ensured in the merging process -- does the system invoke simulators or rely only on syntactic checks?
4. The three challenges listed in the introduction section seem broadly applicable to other coding domains. What aspects are unique to Verilog generation?
5. Could you show per-iteration LocalV results in Fig.5(a)? Basically how does the debugging process in LocalV help improve performance?
6. How does LocalV's retrieval mechanism differ from standard RAG approaches? Any results comparing with RAG?
7. Apart from function correctness, are the implemented IP blocks efficient enough? Have you examined hardware efficiency metrics such as area or timing?
8. Would reasoning-enhanced models like DeepSeek-R1 further improve LocalV's planning or debugging performance?

---

> ### Author Response · Authors · 2025-11-22
> **Response to Reviewer LYBq (1/3)**
>
> Thank you for your valuable feedback. We appreciate your recognition of our contribution that we established a conceptual foundation. Your insights on the limitations, including discussions on the granularity of document partitioning, deeper analysis of the debugging steps, and circuit quality, have guided our improvements. We have added comprehensive experiments, provided further analysis, and revised the paper accordingly. We will address your concerns as follows:
>
> > **W1. The granularity of document partitioning and task decomposition is unclear.**
>
> We appreciate this question regarding the granularity of our decomposition strategy.
>
> **Implementation of partitioning.** In our current implementation, the partitioning mechanism is intentionally simple yet effective, taking advantage of the inherent modularity typically present in IP-level specification documents. We segment the document according to its semantic section markers (e.g., “##” in Markdown), and each resulting segment is treated as an independent information chunk.
>
> **Our method is robust and can maintain coherence across related fragments.** The decomposition process does not assume that tasks must rely on mutually independent information blocks. Instead, for any given sub-task (such as generating a specific module), our retrieval module automatically identifies and gathers all relevant paragraphs or sections across the entire specification. Overlap between the information retrieved for different sub-tasks is both expected and beneficial. This design ensures that each code-generation step remains coherent and has access to all necessary details, regardless of how those details are distributed throughout the document. For example, in the `e203_ifu` circuit, the two submodules `e203_ifu_ifetch` and `e203_ifu_ift2icb` are each described across two to three separate paragraphs of the document, and all of these descriptions are fully retrieved by our framework, enabling the successful generation of `e203_ifu`.
>
> ---
> > **W2. Without stronger guarantees of merging correctness or synthesized design quality (e.g., timing, area), it is difficult to assess how close LocalV comes to generating deployable industrial RTL.**
>
> We appreciate this question! We address this question from two perspectives: correctness assurance and design quality.
> - Correctness assurance. We verify the merging and debugging stages using the simulator's error feedback. The detailed procedure is as follows:
> 1. The merge agent outputs a piece of candidate Verilog code.
> 2. We run a simulation to obtain a waveform and errors.
> 3. We localize faulty signals by tracing their drivers and dependency chains in AST.
> 4. We pass {AST guidance, simulator error logs, error code regions} to the retriever agent, which returns relevant document sections.
> 5. We provide {AST guidance, simulator errors, error code, retrieved sections} to the debug agent, which outputs an insert/delete action.
> 6. We apply the action to the code and repeat the loop until tests pass or the budget is exhausted.
>
> - Design quality. We use Yosys for synthesis to obtain area results, and OpenSTA to obtain the critical path delay and total power. We then compare these metrics with the PPA of the golden RTL (from OpenCores, as reported by RealBench). The results for each module have been added to Appendix E of the manuscript, and the average improvements are summarized in the table below. It is shown that our generated modules are very close to the OpenCores golden RTL, with slightly better area and delay in some cases. This demonstrates that our automatically generated designs achieve good quality.
>
> | |Area improvement|Delay improvement|Power improvement|
> |-|-|-|-|
> |Ours|0.25% |3.00%|-1.15%|
>
> ---
> > **Q1. Does partition granularity impact performance? How does LocalV ensure coherence when multiple paragraphs describe interdependent modules?**
>
> We appreciate this question! Please refer to W1 for our response. Additionally, we conducted a new experiment to show the partition granularity impact performance. We deliberately damaged the document's information locality by randomly combining paragraphs (i.e., we made the document "flatten"). Specifically, we conduct experiments on RealBench by first setting ratios and then combining paragraphs to make the total number of paragraphs to that certain ratio. Then, we use the same workflow of LocalV to generate RTL based on these modified documents. The results are shown in the table below:
>
> | Ratio | Accuracy |
> | - | - |
> | 1x (ours) | 45.0% |
> | 0.8x | 43.3% |
> | 0.7x | 38.3% |
>
> The results show that even when the document is flattened to 0.8, our method still maintains good accuracy. This is because we allow some flexibility during document retrieval, enabling us to handle documents that are not well-structured to some extent. When the document is excessively flattened, such as down to 0.7, the performance drops somewhat, but it still remains 16.7% higher than the baseline MAGE.

---

> ### Author Response · Authors · 2025-11-22
> **Response to Reviewer LYBq (2/3)**
>
> > **Q2. Are the extracted paragraphs validated as meaningful semantic units, or could irrelevant segments mislead sub-task generation?**
>
> We appreciate this question! Yes, while we filter obvious noise (e.g., images in specs), occasional irrelevant segments may still be retrieved. However, in practice, this has limited impact because we allow some flexibility during document retrieval, such as retrieving multiple document fragments to implement a single piece of code.
>
> Additionally, we conducted a new experiment to show that the retrieval of corresponding paragraphs is accurate. Specifically, we manually annotated the golden document fragments that should be retrieved for the AES IP in RealBench and used them as a test set to evaluate retrieval quality. The results in the table below show that both LocalV (Claude) and LocalV (DeepSeek-V3) achieve high retrieval accuracy.
>
> | |Precision|Recall|
> |-|-|-|
> |LocalV (DeepSeek-V3)|0.93|0.93|
> |LocalV (Claude)|0.92|0.95|
>
> ---
> > **Q3. How is correctness ensured in the merging process -- does the system invoke simulators or rely only on syntactic checks?**
>
> Thank you for raising this concern. Yes, LocalV uses simulators to get the error feedback and debug the design accordingly. We have provided the detailed explanation in W2.
>
> ---
> > **Q4. The three challenges listed in the introduction section seem broadly applicable to other coding domains. What aspects are unique to Verilog generation?**
>
> We appreciate this question! We explain the differences between hardware and software coding domains from two aspects: LLMs' capabilities and the inherent characteristics of the tasks.
> - **LLMs' capabilities.** LLMs are significantly less capable at generating hardware description language (HDL) code. Their proficiency in Verilog is substantially weaker. For example, on VerilogEval, an entry-level Verilog generation benchmark, even a model like GPT-4o achieves only 60.1% Pass@1 accuracy [1]. In contrast, on a Python benchmark of comparable difficulty, HumanEval, GPT-4o reaches 92.7% accuracy [2]. This limitation in fundamental model capability makes the associated challenges even harder.
> - **Task characteristics.** As illustrated in Figure 3 of our paper, HDL code generation exhibits much stronger information locality than coding tasks in software (i.e., comparing the E203 CPU top module with the Parse Lisp Expression problem). Based on this hypothesis, we construct our entire framework to alleviate the three challenges.
>
> [1] CodeV-R1: Reasoning-Enhanced Verilog Generation.
>
> [2] EvalPlus Leaderboard. https://evalplus.github.io/leaderboard.
>
> ---
> > **Q5. Could you show per-iteration LocalV results in Fig.5(a)?**
>
> We appreciate this question! We have added the accuracy progression during the debugging process, as shown in the table below. It is shown that the accuracy increases steadily as debugging proceeds and eventually levels off. As stated in the previous question, the debugging process corrects erroneous RTL code by using feedback from the simulator.
>
> |Iteration|Accuracy|
> | - | - |
> | 0 | 29.00% |
> | 1 | 35.33% |
> | 2 | 36.67% |
> | 3 | 38.67% |
> | 4 | 39.00% |
> | 5 | 39.33% |
> | 6 | 40.33% |
> | 7 | 41.00% |
> | 8 | 41.33% |
> | 9 | 41.33% |
> | 10 | 42.33% |
>
> ---
> > **Q6. How does LocalV's retrieval mechanism differ from standard RAG approaches? Any results comparing with RAG?**
>
> We appreciate this question!
>
> Traditional RAG methods typically perform retrieval by computing similarity scores using an embedding model. However, (1) this approach is heavily dependent on the capability of the embedding model, (2) it struggles to effectively analyze and reason about design-related information—such as simulator error feedback, and (3) it is difficult to determine an appropriate number of retrieved documents, making it inflexible when multiple document segments correspond to multiple code snippets. In contrast, our method leverages the LLM itself to perform the retrieval, enabling it to effectively handle the issues above and thus achieve better performance on complex tasks such as IP-level design.
>
> We conducted new experiments on the retrieval stage of debugging (since we cannot replace the planning stage with RAG), using two common embedding models, Qwen3-Embedding-0.6B and Qwen3-Embedding-8B, with the results shown in the table below. It is shown that our method offers a clear advantage over embedding-based RAG. Note that the embedding model shows identical precision and recall because we set the number of retrieved items to match the ground truth.
>
> | |Precision|Recall|
> |-|-|-|
> |Qwen3-Embedding-0.6B|48.21%|48.21%|
> |Qwen3-Embedding-8B|56.99%|56.99%|
> |Ours|95.57%|86.16%|

---

> ### Author Response · Authors · 2025-11-22
> **Response to Reviewer LYBq (3/3)**
>
> > **Q7. Apart from function correctness, are the implemented IP blocks efficient enough? Have you examined hardware efficiency metrics such as area or timing?**
>
> We appreciate this question! As shown in Weakness 2, we have evaluated the PPA of the generated RTL, and our generated modules are very close to the OpenCores golden RTL, with slightly better area and delay in some cases. This demonstrates that our automatically generated designs achieve good quality.
>
> ---
> > **Q8. Would reasoning-enhanced models like DeepSeek-R1 further improve LocalV's planning or debugging performance?**
>
> We appreciate this question! While reasoning-enhanced models are promising, we found that applying DeepSeek-R1 to the challenging RealBench benchmark presents significant practical hurdles. specifically:
> 1. Efficiency and Stability: We observed that DeepSeek-R1's average output length is approximately 5–6 times longer than Claude's due to its extensive reasoning chains. This excessive length frequently leads to API timeouts and output formatting errors, hindering the planning and debugging loop.
> 2. Performance: Preliminary tests on the SDC subset of RealBench show that DeepSeek-R1 achieves a success rate of only 7% by a low workflow success rate, significantly lower than the 35% achieved by LocalV with Claude.
> Given these trade-offs, we conclude that DeepSeek-R1 is not currently an efficient or effective choice for this specific setting compared to our current backbone.
>
> ---
> Thank you again for your valuable feedback. We hope the rebuttal is adequate in addressing your concerns, and if so, please update your review in accordance. If any questions remain, we welcome further discussion.

---

> > ### Comment · Reviewer_LYBq · 2025-11-28
> >
> > Thank you for the detailed response. The authors addressed most of my questions. However, based on the reported results, it appears that in most cases the PPA metrics remain unchanged. How can the authors ensure that the LLM is not simply recalling the original code rather than performing meaningful optimization?
> >
> > Moreover, the results in Table 5 suggest that the benchmark may be too easy for current LLMs: no further improvements are observed, and many designs seem to have reached a local minimum. This raises concerns about whether LocalV actually provides measurable benefits to LLM performance, or whether the evaluation setting is insufficiently challenging to demonstrate such improvements.

---

> > > ### Author Response · Authors · 2025-12-02
> > >
> > > Thank you for your response. In this paper, we focus more on code correctness rather than PPA optimization, because correctness in IP-level module design is a prerequisite for PPA optimization and remains an unresolved challenge. PPA optimization of modules is a separate research direction; we conducted the experiment only to demonstrate that even without dedicated PPA tuning, LocalV can still produce circuits with acceptable PPA.
> > > > **How can the authors ensure that the LLM is not simply recalling the original code rather than performing meaningful optimization?**
> > >
> > > We address this from two perspectives to show that the LLM is not simply recalling the original code:
> > > - **Low Code Similarity:** After removing comments, identical module headers and syntactic elements (e.g., `begin`), **only 14% of the code lines were identical.** Many of these are commonly repeated lines, such as `reg delay;` or port mappings like `.addr(addr),`.
> > > - **Performance Gap:** The baseline LLM performs **25% worse in accuracy** than our approach, indicating that the model itself has not memorized relevant code knowledge, it is our framework that enhances its correctness.
> > >
> > > We also found that, some different verilog implementations are synthesized into netlists with the same structure. In Appendix E, we have added an explanation regarding this phenomenon of identical PPA: We present a substructure from the code generated by LocalV and the golden code for the `e203_exu_disp` module. Although these two implementations totally differ, they were synthesized into the same netlist structure, resulting in identical PPA metrics. The complete code is provided in Appendix E.
> > >
> > > ```verilog
> > > // Golden
> > > `include "e203_defines.v"
> > > module top(
> > >     input  oitf_empty,
> > >     input  wfi_halt_exu_req,
> > >     input  disp_oitf_ready,
> > >     input  [`E203_DECINFO_WIDTH-1:0] disp_i_info,
> > >     input  oitfrd_match_disprd,
> > >     input  oitfrd_match_disprs1,
> > >     input  oitfrd_match_disprs2,
> > >     input  oitfrd_match_disprs3,
> > >     output disp_condition
> > > );
> > > wire [`E203_DECINFO_GRP_WIDTH-1:0] disp_i_info_grp = disp_i_info [`E203_DECINFO_GRP];
> > > wire disp_csr = (disp_i_info_grp == `E203_DECINFO_GRP_CSR);
> > > wire disp_alu_longp_prdt = (disp_i_info_grp == `E203_DECINFO_GRP_AGU);
> > > wire disp_fence_fencei = (disp_i_info_grp == `E203_DECINFO_GRP_BJP) &
> > >                         (disp_i_info [`E203_DECINFO_BJP_FENCE] |
> > >                         disp_i_info [`E203_DECINFO_BJP_FENCEI]);
> > > wire raw_dep =  ((oitfrd_match_disprs1) | (oitfrd_match_disprs2) | (oitfrd_match_disprs3));
> > > wire waw_dep = (oitfrd_match_disprd);
> > > wire dep = raw_dep | waw_dep;
> > > wire disp_condition = (disp_csr ? oitf_empty : 1'b1)
> > >                     & (disp_fence_fencei ? oitf_empty : 1'b1)
> > >                     & (~wfi_halt_exu_req)
> > >                     & (~dep)
> > >                     & (disp_alu_longp_prdt ? disp_oitf_ready : 1'b1);
> > > endmodule
> > > ```
> > >
> > > ```verilog
> > > // LocalV
> > > `include "e203_defines.v"
> > > module top(
> > >     input  oitf_empty,
> > >     input  wfi_halt_exu_req,
> > >     input  disp_oitf_ready,
> > >     input  [`E203_DECINFO_WIDTH-1:0] disp_i_info,
> > >     input  oitfrd_match_disprd,
> > >     input  oitfrd_match_disprs1,
> > >     input  oitfrd_match_disprs2,
> > >     input  oitfrd_match_disprs3,
> > >     output disp_condition
> > > );
> > >     wire disp_csr  = (disp_i_info[`E203_DECINFO_GRP] == `E203_DECINFO_GRP_CSR);
> > >     wire disp_agu  = (disp_i_info[`E203_DECINFO_GRP] == `E203_DECINFO_GRP_AGU);
> > >     wire disp_bjp  = (disp_i_info[`E203_DECINFO_GRP] == `E203_DECINFO_GRP_BJP);
> > >     wire disp_fence  = disp_bjp & disp_i_info[`E203_DECINFO_BJP_FENCE];
> > >     wire disp_fencei = disp_bjp & disp_i_info[`E203_DECINFO_BJP_FENCEI];
> > >     wire need_wait_oitf_empty = disp_csr | disp_fence | disp_fencei;
> > >     wire rs1_dep_oitf_idx = oitfrd_match_disprs1;
> > >     wire rs2_dep_oitf_idx = oitfrd_match_disprs2;
> > >     wire rs3_dep_oitf_idx = oitfrd_match_disprs3;
> > >     wire disp_raw_dep = rs1_dep_oitf_idx | rs2_dep_oitf_idx | rs3_dep_oitf_idx;
> > >     wire disp_waw_dep = oitfrd_match_disprd;
> > >     wire oitf_empty_condition = (~need_wait_oitf_empty) | (need_wait_oitf_empty & oitf_empty);
> > >     wire no_dep_condition = (~disp_raw_dep) & (~disp_waw_dep);
> > >     wire wfi_halt_condition = ~wfi_halt_exu_req;
> > >     wire agu_oitf_ready = (~disp_agu) | (disp_agu & disp_oitf_ready);
> > >     wire disp_condition = oitf_empty_condition & wfi_halt_condition & no_dep_condition & agu_oitf_ready;
> > > endmodule
> > > ```
> > >
> > > > **This raises concerns about whether LocalV actually provides measurable benefits to LLM performance**
> > >
> > > As mentioned, our main contribution lies in improving correctness, a prerequisite for PPA optimization. According to Table 2, our method substantially improves correctness over the baseline (by 25%). Moreover, Table 5 shows that the PPA of the auto-generated designs is comparable to that of human-written golden code, demonstrating their practical usability. In the future, we plan to explore PPA optimization specifically for IP-level module design, this would involve different techniques and represents a new research challenge.

---

### Official Review · Reviewer_aBM8 · 2025-10-31

**Soundness:** 3
**Presentation:** 3
**Contribution:** 3
**Rating:** 4
**Confidence:** 3

**Summary:**

This paper tackles the scalability bottleneck of large-language-model (LLM)–based RTL code generation.
Existing approaches struggle with IP-level (industrial-scale) designs because specifications are long, code is lengthy and inter-dependent, and debugging is hard.
The authors formalize the Information Locality principle — the assumption that each RTL semantic unit depends mainly on a small, localized portion of the specification. They quantify locality using a normalized-entropy metric based on embedding similarity and show that hardware specifications display stronger locality than software tasks.
Building on this, they propose LocalV, a five-stage multi-agent pipeline: hierarchical document indexing; task planning and skeleton generation; localized RTL generation; interface-consistent merging; AST-guided locality-aware debugging.
On the REALBENCH benchmark (AES, SDC, E203 CPU), LocalV achieves 75 % syntax and 45 % functional Pass@1 — about 23 % higher than the best agent baseline (MAGE).

**Strengths:**

1.	Quantitative validation of Information Locality (Eqs. 1–4, Fig. 3) is a novel and principled idea beyond heuristic agent planning.
2.	Dual-level indexing, deterministic planning, and AST-guided debugging form a coherent, effective pipeline.
3.	Strong improvement on REALBENCH over model and agent baselines.

**Weaknesses:**

1.	The framework's core assumption is the availability of a complete, detailed, and well-structured natural-language specification as input. This is a significant prerequisite that defines a more constrained problem setting than what many general-purpose agent systems address. In practice, such a document may not exist and would need to be generated from higher-level requirements, a task not covered by the proposed workflow. The paper should discuss this dependency, as the method's performance is tightly coupled to the quality of this assumed input.
2.	The Information Locality hypothesis is validated on a small set (one CPU and a handful of modules). The reported entropy gaps are modest, and the analysis lacks confidence intervals or sensitivity analyses for the embedding model used. The external validity across more IPs and different document styles (e.g., less structured or "flat" specifications) remains unsubstantiated.
3.	The pipeline depends critically on the initial indexing and planning stages, yet there are no metrics for the accuracy of these steps or an analysis of how early mistakes propagate or are corrected. The current ablations only report final outcomes, while insights into process-level robustness are missing.
4.	The ablation study shows large gains from debugging, but the paper fixes a 10-iteration debug cap without justification. There is no analysis of the cost-benefit trade-off (e.g., a Pass@k vs. iterations curve) or an equal-budget comparison against baselines. This makes it difficult to distinguish whether the performance advantage stems from methodological superiority or simply a larger budget for repair and search.
5.	The debug agent lacks a reproducible algorithmic description. While the paper claims "AST-guided, locality-aware" debugging, it does not provide a clear workflow, localization hit-rates, or controlled ablations against simpler debugging strategies (e.g., without retrieval or without AST guidance) under the same budget.
6.	Key details regarding efficiency and resource usage are missing, such as end-to-end runtime, total token consumption, and the complexity of the merging stage. The stopping criteria for the debugging loop are also unspecified.
7.	While the evaluation on REALBENCH is well-aligned with the paper's goals, its scope is narrow. The findings could be further strengthened by demonstrating performance on a broader set of benchmarks—such as HDLBench for larger SoCs and the relevant subsets of CVDP for agent-level tool integration—to provide additional insights into the method's scalability and robustness.
8.	Reproducibility is limited by the lack of variance reporting over multiple runs, full baseline configurations (e.g., temperature, sampling counts), and public access to data splits or evaluation scripts.
9.	The mechanism for ensuring interface consistency during the merging stage is not formalized. It is unclear how potential conflicts (e.g., macros, parameters) are resolved or what the computational complexity of this step is for larger designs

**Questions:**

1.	Regarding the baseline comparisons in Table 1, could you provide more details to justify the fairness of the evaluation? Specifically, can you clarify the resource budgets (e.g., total tokens, time, or iterations) allocated to LocalV versus MAGE/VerilogCoder? This would help distinguish the methodological advantages from potential differences in computational effort.
2.	The concept of 'information locality' is central to the paper. Could you elaborate on the characteristics of a specification document that are necessary for this principle to hold? For instance, how detailed or hierarchically structured must the document be for the locality to be strong? A discussion on the boundary conditions or the types of specifications where this principle might weaken would be valuable.
3.	For the debugging stage, could you provide a Pass@1 vs. #iterations curve (from 0 to 10) along with the associated time/token costs? This would help clarify the cost-benefit trade-off of the iterative repair process.
4.	Could you provide a more detailed, reproducible workflow (e.g., pseudocode) for the Debug Agent? It would be helpful to understand the distinct roles of AST analysis and locality-aware retrieval in the error localization and patching process.
5.	What is the "localization hit rate" of the debug agent? Specifically, what percentage of the time does it correctly map a simulation error to the responsible code fragment and, crucially, to the correct specification fragment?
6.	Could you provide accuracy/recall metrics for the initial hierarchical indexing and skeleton planning stages? It would be useful to understand how robust these critical early steps are.
7. Others see Weakness.

---

> ### Author Response · Authors · 2025-11-22
> **Response to Reviewer aBM8 (1/4)**
>
> Thank you for your valuable feedback. We appreciate your recognition of our contributions, including the novel and principled and strong improvement. Your insights on the limitations, including discussions on the core assumption, deeper analysis of the retrieving and debugging steps, cost efficiency, and more evaluations needed, have guided our improvements. We have added comprehensive experiments, provided further analysis, and revised the paper accordingly. We will address your concerns as follows:
>
> > **W1. In practice, such a document may not exist and would need to be generated from higher-level requirements, a task not covered by the proposed workflow. The paper should discuss this dependency, as the method's performance is tightly coupled to the quality of this assumed input.**
>
> Thank you for concerning the locality assumption of LocalV. We acknowledge the reliance on structured specifications and will explicitly discuss this dependency in the revised paper. We argue that in the context of **IP-level hardware design**, this assumption reflects standard industry practice rather than an artificial constraint. Unlike general software development, high-quality hardware IPs typically demand rigorous specifications (e.g., datasheets) *before implementation*. These documents are inherently written for human readability, naturally organizing sub-modules, interfaces, and instantiation logic into distinct sections. Consequently, LocalV's requirement for well-structured input aligns with the reality of the hardware design workflow.
> To quantitatively validate the impact of this inherent structure, we deliberately damaged the document information locality by randomly combining paragraphs (i.e., we made the document "flatten"). Specifically, we conduct experiments on RealBench by first setting ratios and then combining paragraphs to make the total number of paragraphs to that certain ratio. Then, we use the same workflow of LocalV to generate RTL based on these modified documents. The results are shown in the table below:
>
> | Ratio | Accuracy |
> | - | - |
> | 1x (ours) | 45.0% |
> | 0.8x | 43.3% |
> | 0.7x | 38.3% |
>
> The results show that even when the document is flattened to 0.8, our method still maintains good accuracy. This is because we allow some flexibility during document retrieval, enabling us to handle documents that are not well-structured to some extent. When the document is excessively flattened, such as down to 0.7, the performance drops somewhat, but it still remains 16.7% higher than the baseline MAGE.
>
> ---
> > **W2. The Information Locality hypothesis is validated on a small set (one CPU and a handful of modules). The reported entropy gaps are modest, and the analysis lacks confidence intervals or sensitivity analyses for the embedding model used.**
>
> Thank you for pointing this out. We have strengthened the validation of the information locality hypothesis through extended experiments and added them to the paper's main text.
> To address **embedding model sensitivity** concerns, we cross-validated our metrics using an RTL embedding model DeepRTL2 [1], a larger embedding model Qwen3-8B and a lexical method BM25. For **more IPs**, we emphasize that we have tested **the whole RealBench** in the end of Section 3.2 of our paper, and we add the results for RealBench under different embedding models. For **different document styles**, we add the results by splitting the document to different number of fragments. As shown in the table below, strong locality signals are consistently observed across different models and algorithms, confirming that the locality is a property of the data, not an artifact of the embedding model.
>
> | Module | BM25 | DeepRTL2 | Qwen-0.6B | Qwen-8B |
> | :--- | :--- | :--- | :--- | :--- |
> | E203 CPU Top Module | 0.4036 | 0.9504 | 0.8680 | 0.8873 |
> | E203 CPU (0.8x Doc) | 0.3284 | 0.9461 | 0.8839 | 0.9045 |
> | E203 CPU (0.6x Doc) | 0.3224 | 0.9552 | 0.8806 | 0.8902 |
> | Parse Lisp Expression | 0.5412 | 0.9699 | 0.9126 | 0.9453 |
> | 10 Combined Modules | 0.0669 | 0.9255 | 0.8206 | 0.8167 |
> | RealBench Average | 0.2083 | 0.9231 | 0.8406 | 0.8607 |
>
> The results have shown that, for the whole RealBench, the information locality is consistently higher (under different embedding models) than the software problem "Parse Lisp Expression" and generally lower than "10 Combined Modules". In addition, different document styles (different segmentation of the E203 CPU Top Module) do not change the locality much.
>
> [1] DeepRTL2: A Versatile Model for RTL-Related Tasks

---

> ### Author Response · Authors · 2025-11-22
> **Response to Reviewer aBM8 (2/4)**
>
> > **W3. The pipeline depends critically on the initial indexing and planning stages, yet there are no metrics for the accuracy of these steps or an analysis of how early mistakes propagate or are corrected.**
>
> Thank you for your concern on indexing and planning stages. We assessed LocalV’s sensitivity to the indexing and retrieval components through two key ablation studies:
>
> 1.**Necessity.** The retrieval step is essential: removing the index entirely led to a performance drop **from 45% to 35%**.
>
> 2.**Robustness.** The system is highly robust to the choice of backbone model. We substituted the primary planner with DeepSeek-V3 still achieved a comparable 41.6% pass rate. We attribute this robustness to our dual-level description strategy, which renders the indexing task well-defined and tractable. To further validate this, we manually annotated the golden document fragments that should be retrieved for the AES IP in RealBench and used them as a test set to evaluate retrieval quality. The results in the table below show that both LocalV (Claude) and LocalV (DeepSeek-V3) achieve high retrieval accuracy. This demonstrates that the indexing step is not a performance bottleneck, as our design supports consistent retrieval quality across models with different capabilities.
>
> | |Precision|Recall|
> |-|-|-|
> |LocalV (DeepSeek-V3)|0.93|0.93|
> |LocalV (Claude)|0.92|0.95|
>
> ---
> > **W4. There is no analysis of the cost-benefit trade-off (e.g., a Pass@k vs. iterations curve) or an equal-budget comparison against baselines.**
>
> Thank you for raising this question. Our primary intention to use 10 iterations is to guarantee that the iteration is sufficient.
> To investigate the relation between the number of iterations and the final accuracy as well as token cost, we run 10 debug iterations 5 times and average the accuracy at iterations 0~10 over these 5 trials, added in Appendix D. The statistics are shown in the table below:
>
> | Iteration | Accuracy | Token Count |
> | - | - | - |
> | 0 | 29.00% | 66,058 |
> | 1 | 35.33% | 83,386 |
> | 2 | 36.67% | 99,948 |
> | 3 | 38.67% | 115,975 |
> | 4 | 39.00% | 132,069 |
> | 5 | 39.33% | 147,885 |
> | 6 | 40.33% | 163,635 |
> | 7 | 41.00% | 179,543 |
> | 8 | 41.33% | 194,791 |
> | 9 | 41.33% | 209,998 |
> | 10 | 42.33% | 224,978 |
>
> This experiment shows that although we use 10 iterations in our main experiment, debugging for fewer iterations (for example, 5 iterations) costs fewer tokens but can still achieve good results. We also compare LocalV with MAGE under a comparable token budget in the following table:
>
> | | RealBench Acc | Avg Input Token | Avg Output Token | Avg Total Token |
> | - | - | - | - | - |
> | MAGE | 21.6% | 180,712 | 38,064 | 218,776 |
> | LocalV | 42.33% | 201,961 | 23,017 | 224,978 |
>
> Under comparable token budgets (although LocalV costs more total tokens, it generates less costly output tokens, making a slightly lower total cost in general), LocalV still significantly outperforms MAGE, indicating gains arise from our method rather than a larger repair/search budget.
> We include an accuracy vs. iterations plot and per-iteration token usage in the appendix to make the trade-off explicit.
>
> ---
> > **W5. The debug agent lacks a reproducible algorithmic description.**
>
> Thanks for the suggestion. We have added pseudocode of the debug workflow in the appendix for reproducibility. The workflow is summarized as follows:
> 1. The merge agent outputs a piece of candidate Verilog code.
> 2. We run a simulation to obtain a waveform and errors.
> 3. We localize faulty signals by tracing their drivers and dependency chains in AST.
> 4. We pass {AST guidance, simulator error logs, error code regions} to the retriever agent, which returns relevant document sections.
> 5. We provide {AST guidance, simulator errors, error code, retrieved sections} to the debug agent, which outputs an insert/delete action.
> 6. We apply the action to the code and repeat the loop until tests pass or the budget is exhausted.
> We have conducted ablations for AST guidance and retrieval in the debug loop. The results are shown in the table below:
>
> | Setting | Accuracy |
> | -- | -- |
> | LocalV | 45.0% |
> | w/o AST Guidance | 40.0% |
> | w/o Retrieval | 38.3% |
>
> The results have shown that both AST guidance and retrieval play important roles in the debug loop.
> Also, we manually annotated the golden document fragments that should be retrieved during debugging. Precision and recall for document retrieval were calculated using the standard definitions, against this golden set of annotated relevant fragments. The results are shown in the table below:
>
> | |Precision|Recall|
> |-|-|-|
> |LocalV (Claude)|0.95|0.86|
>
> We can see the retriever's localization hit-rate is high with a Precision/Recall of 0.95 / 0.86, which attributes to our **dual-level description strategy** in document section indexing.

---

> ### Author Response · Authors · 2025-11-22
> **Response to Reviewer aBM8 (3/4)**
>
> > **W6. Key details regarding efficiency and resource usage are missing, such as end-to-end runtime, total token consumption, and the complexity of the merging stage. The stopping criteria for the debugging loop are also unspecified.**
>
> Thank you for your valuable suggestion. For total token consumption, we have provided detailed statistics in the answer for W4.
> For the complexity of the merging stage, we leave it to W9.
> For the stopping criteria of the debug loop, it terminates when 1) all unit tests pass or 2) max iterations reached (default 10).
>
> ---
> > **W7. The findings could be further strengthened by demonstrating performance on a broader set of benchmarks—such as HDLBench for larger SoCs and the relevant subsets of CVDP for agent-level tool integration—to provide additional insights into the method's scalability and robustness.**
>
> We appreciate the reviewer’s suggestion to broaden the evaluation scope.
>
> Although we were unable to find HDLBench, we evaluated LocalV on the CVDP [1] benchmark, specifically the non-agentic part of `cid003` suite (78 Spec-to-RTL problems). We exclude the agentic part since it requires abilities other than RTL generation, such as reading and writing files via the command line, and navigating, organizing, and pinpointing issues across multiple code files. These requirements are beyond the topic of LocalV (IP-level spec-to-rtl), and are orthogonal to LocalV's agent ability. Following CVDP, we evaluated pass@1 accuracy. The results are shown in the following table.
>
> | | CVDP Accuracy |
> | - | - |
> | Claude-3.7 | 48.72% |
> | MAGE (Claude) | 44.87% |
> | Ours (Claude) | 61.50% |
>
> While these tasks have relatively short contexts (avg. ~1,100 tokens) and thus do not fully demonstrate LocalV’s long-context ability, LocalV still significantly outperforms direct sampling and MAGE, showing strong robustness. These results indicate that LocalV maintains superior performance across tasks with different specification styles, supporting its scalability and robustness beyond Realbench.
>
> [1] Comprehensive Verilog Design Problems: A Next-Generation Benchmark Dataset for Evaluating Large Language Models and Agents on RTL Design and Verification
>
> ---
> > **W8. Reproducibility is limited by the lack of variance reporting over multiple runs, full baseline configurations (e.g., temperature, sampling counts), and public access to data splits or evaluation scripts.**
>
> Thank you for the advice of running multiple times. We conducted 5 independent runs with different seeds, and the results are shown in the following table:
>
> | Run Number | 1 | 2 | 3 | 4 | 5 |
> | - | - | - | - | - | - |
> | Accuracy | 40.0% | 41.7% | 45.0% | 43.3% | 41.7% |
>
> The mean is 42.3% and the standard deviation is 1.7%
> Also, we provide the configurations for LocalV and MAGE (baseline) here for exact replication:
> For LocalV, we use temperature = 0.6，top-p = 1.0 during rollout.
> For MAGE, we use its default parameter, temperature = 0.85 and top-p = 0.95. Besides, for a comparable token-budget comparison, we restrict its RTL candidate count to 2. We have also run an additional experiment using the same temperature (0.6) and top-p (1.0), MAGE achieves 23.3% accuracy, still far below LocalV.
>
> We will release the complete LocalV implementation as an open-source repository upon acceptance.
>
> ---
> > **W9. The mechanism for ensuring interface consistency during the merging stage is not formalized. It is unclear how potential conflicts (e.g., macros, parameters) are resolved or what the computational complexity of this step is for larger designs.**
>
> Thank you for pointing this out. We should emphasize that the interface consistency during the merging stage is **handled by LLMs instead of predefined rules**, which allows it to leverage semantic understanding and context for superior flexibility and adaptability. The Merge Agent and the Debug Agent are responsible for this stage.
>
> When merging decomposed code modules, the Merge Agent is responsible for the initial integration. It is designed to identify and resolve potential conflicts, such as macro redefinitions or parameter mismatches. If any subtle inconsistencies or bugs persist after this step, the Debug Agent is subsequently invoked to perform a more in-depth analysis and apply necessary corrections.
> Regarding the computational complexity for larger designs, we use the token count as a proxy. We have to emphasize that the primary bottleneck, code generation, has been effectively addressed through decomposing into individual modules, each exhibiting sub-linear complexity. While we acknowledge that the current Merge Agent scales linearly in token usage with total code length, this step is inherently less complex than code generation, as it primarily involves the integration of already implemented modules. Exploring more scalable merging strategies that further reduce contextual dependency remains a key direction for our future work.

---

> ### Author Response · Authors · 2025-11-22
> **Response to Reviewer aBM8 (4/4)**
>
> > **Q1. Regarding the baseline comparisons in Table 1, could you provide more details to justify the fairness of the evaluation?**
>
> Thank you for your suggestion. Please refer to our answer to W4 for the equal token-budget comparison.
>
> ---
> > **Q2. Could you elaborate on the characteristics of a specification document that are necessary for this principle to hold?**
>
> We thank the reviewer for raising this point. We think the specification should ideally exhibit characteristics such as:
> 1.**Modularity:** The document is naturally divided into sections or chapters that correspond to distinct functional blocks or sub-modules (e.g., a dedicated section for a memory controller, another for a data processing pipeline).
>
> 2.**Self-Containment:** Within a module's section, the description of its interface (inputs/outputs), functionality, and key internal logic is highly concentrated. An agent can understand and generate the code for that module by primarily reading its dedicated section, with minimal need to cross-reference distant parts of the document.
>
> 3.**Clear Interface Contract:** The interactions between modules are explicitly defined through well-specified interfaces (e.g., "the FIFO controller provides a data_valid signal to the processing unit"). This confines the "integration knowledge" to specific, manageable points.
>
> For boundary conditions that may weaken this principle, below are some scenarios:
> 1. Tightly-Coupled and Cross-Cutting Concerns: Specifications for systems with complex, global state management or pervasive logic (e.g., a sophisticated interrupt controller where many modules can assert interrupts) require constant cross-referencing, violating self-containment.
> 2. Non-Modular or "Flat" Documents: A single, continuous narrative of requirements without clear hierarchical separation (e.g., "The system shall first do A, then B, and if condition C is met, it must update global state D and notify module E") forces an agent to maintain a large context.
> We have provided two examples, a positive example in RealBench and a negative example of the "Parse Lisp Expression" in Appendix F.
>
> ---
> > **Q3. For the debugging stage, could you provide a Pass@1 vs. #iterations curve (from 0 to 10) along with the associated time/token costs?**
>
> Thank you for this valuable suggestion. To clarify the cost-benefit trade-off of our iterative debugging process, we have plotted Pass@1 versus the number of debug iterations (from 0 to 10) and included it in the appendix. The details are provided in W4.
>
> ---
> > **Q4. Could you provide a more detailed, reproducible workflow (e.g., pseudocode) for the Debug Agent?**
>
> Thanks for the suggestion. We have provided details in response to W5.
>
> ---
> > **Q5. What is the "localization hit rate" of the debug agent?**
>
> Thank you for your question. We have explained the hit rate in W5.
>
> ---
> > **Q6. Could you provide accuracy/recall metrics for the initial hierarchical indexing and skeleton planning stages?**
>
> Thank you for raising this point. We have provided the detailed precision and recall statistics in our response to W3, which we believe will fully address your inquiry.
>
> ---
> Thank you again for your valuable feedback. We hope the rebuttal is adequate in addressing your concerns, and if so, please update your review in accordance. If any questions remain, we welcome further discussion.

---

### Official Review · Reviewer_D3Dj · 2025-10-31

**Soundness:** 3
**Presentation:** 3
**Contribution:** 2
**Rating:** 4
**Confidence:** 4

**Summary:**

This paper proposes LocalV, a multi-agent framework for IP-level Verilog generation. The authors identify three fundamental challenges in generating IP-level Verilog code: long-document handling, long-code generation, and the complex debugging process. To address these challenges, LocalV leverages the inherent information locality in modular hardware design. The framework incorporates three key components: an index-driven document partitioning mechanism, a fragment-based generation strategy that decomposes complex tasks into manageable subtasks, and a traceable debugging pipeline that maps errors back to relevant specification fragments through AST-guided analysis. LocalV achieves a 45.0% pass rate on RealBench, surpassing state-of-the-art agent-based frameworks by 23.4%.

**Strengths:**

This paper addresses an underexplored problem: generating functionally correct Verilog code for large-scale, IP-level designs using large language models (LLMs). The proposed LocalV framework builds on the insightful observation of *information locality* in hardware specifications, demonstrating that modular design documents can be decomposed into semantically cohesive segments for localized code generation. The authors provide empirical justification for this hypothesis through an entropy-based locality analysis and integrate it into a multi-agent architecture comprising document partitioning, planning, localized generation, merging, and debugging. Experimental results on the challenging RealBench dataset show noticeable improvements over previous baselines, with clear component-level contributions validated by an ablation study. The paper is well written and systematically organized, and the workflow diagrams enhance readability and make the overall system easy to follow.

**Weaknesses:**

Despite its strong motivation, the technical novelty of LocalV is somewhat incremental compared to prior multi-agent systems such as MAGE and VerilogCoder. The main conceptual contribution lies in leveraging information locality to guide task granularity rather than introducing fundamentally new agent coordination mechanisms. The evaluation scope is also somewhat narrow, as RealBench is the only benchmark used. While the results are encouraging, it remains unclear how well the proposed approach generalizes to other hardware design domains or specification styles beyond those captured in RealBench. Moreover, the empirical analysis focuses primarily on Pass@k metrics and lacks a deeper investigation of failure cases. For example, scenarios where LocalV fails to produce functionally correct or syntactically valid code. Such analysis would provide valuable insights into the method’s current limitations and potential directions for improvement. Overall, while the observed gains are promising, they may not fully justify the added complexity of the multi-agent pipeline. The work would benefit from broader evaluation, more transparent implementation details, and stronger empirical evidence supporting the generality of the information-locality hypothesis.

**Questions:**

1. How sensitive is LocalV to the accuracy of the document indexing and retrieval steps?
2. Can LocalV effectively scale to system-level designs exceeding 2,000 lines of RTL code or to multi-module integration scenarios?
3. Will the authors release the LocalV implementation to facilitate reproducibility and community validation?

---

> ### Author Response · Authors · 2025-11-22
> **Response to Reviewer D3Dj (1/2)**
>
> Thank you for your valuable feedback. We appreciate your recognition of our contributions, including the insightful observation, well-organized paper, and the step toward practical IP-level design. Your insights on the limitations, including further comparisons with previous work, analysis of failure cases, and a request for the plan of open-source, have guided our improvements. We have added comprehensive experiments, provided further analysis, and revised the paper accordingly. We will address your concerns as follows:
>
> > **W1. The main conceptual contribution lies in leveraging information locality to guide task granularity rather than introducing fundamentally new agent coordination mechanisms.**
>
> We thank the reviewer for the insightful comparison. In the following table, we compare the workflow of our LocalV with MAGE and VerilogCoder:
>
> | Component | LocalV | VerilogCoder | MAGE |
> | - | - | - | - |
> | Plan Agent | $\checkmark$ | $\checkmark$ | $\times$ |
> | **Indexing Original Document During Planning** | $\checkmark$ | $\times$ | $\times$ |
> | Code Agent | $\checkmark$ | $\checkmark$ | $\checkmark$ |
> | Merge Agent | LLM | Rule | $\times$ |
> | Debug Agent | $\checkmark$ | $\checkmark$ | $\checkmark$ |
> | **Indexing Original Document During Debugging** | $\checkmark$ | $\times$ | $\times$ |
>
> From this table, we can clearly see that MAGE does not have the decomposing step compared to LocalV. Meanwhile, although VerilogCoder has a decompose step, LocalV introduces **original document indexing during both planning and debugging**, and **LLM-based code merging** to address the challenges of **IP-level Verilog generation**.
> For **indexing original document**, it is the core of LocalV to **utilize locality** to address the **excessive context length** problem in complex IP design. In our ablation study (Table 4 in the new paper), removing indexing in planning decreases the functional accuracy **from 45% to 35%**. In addition, we have conducted another ablation study to remove indexing in debugging (please refer to W5 by Reviewer aBM8 for details), and found that it decreases the functional accuracy **from 45% to 38.3%**. From these ablations, it is clear that locality is essential to solve long problems.
> For **LLM-based code merging**, it is more flexible by leaving the module merging to LLM compared to partitioning problems into abstract sub-modules with rigid interfaces in VerilogCoder. The code agent only needs to implement the core Verilog logic (e.g., wiring, independent blocks) without the need of VerilogCoder's Verilog verification assistant to check module syntax.
>
> ---
> > **W2. The evaluation scope is also somewhat narrow, as RealBench is the only benchmark used.**
>
> We appreciate this question! To address the concern regarding evaluation scope, we evaluated LocalV on the CVDP [1] benchmark, specifically the non-agentic part of `cid003` suite (78 Spec-to-RTL problems). We exclude the agentic part since it requires abilities other than RTL generation, such as reading and writing files via the command line, and navigating, organizing, and pinpointing issues across multiple code files. These requirements are beyond the topic of LocalV (IP-level spec-to-rtl), and are orthogonal to LocalV's agent ability. Following CVDP, we evaluated pass@1 accuracy. The results are shown in the following table.
>
> | | CVDP Accuracy |
> | - | - |
> | Claude-3.7 | 48.72% |
> | MAGE (Claude) | 44.87% |
> | Ours (Claude) | 61.50% |
>
> While these tasks have relatively short contexts (avg. ~1,100 tokens) and thus do not fully demonstrate LocalV’s long-context ability, LocalV still significantly outperforms direct sampling and MAGE, showing strong robustness. These results indicate that LocalV maintains superior performance across tasks with different specification styles, supporting its scalability and robustness beyond Realbench.
>
> [1] Comprehensive Verilog Design Problems: A Next-Generation Benchmark Dataset for Evaluating Large Language Models and Agents on RTL Design and Verification
>
> ---
> > **W3. Moreover, the empirical analysis focuses primarily on Pass@k metrics and lacks a deeper investigation of failure cases.**
>
> We appreciate this constructive suggestion. In response, we have conducted a qualitative analysis of failure cases and included a detailed discussion in **Appendix C** of the revised manuscript.
> Our investigation reveals that failures generally stem from three scenarios. First, regarding functional correctness, the model faces difficulties with complex logic, it may fail to generate correct encryption sequences for modules like aes_cipher_top. Second, syntactic errors may occur when handling bit-width definitions that rely on complex macro expressions. Finally, we observed limitations in handling large-scale signals, where an excessive number of signals can overwhelm the model, leading to incorrect signal assignments.

---

> ### Author Response · Authors · 2025-11-22
> **Response to Reviewer D3Dj (2/2)**
>
> > **Q1. How sensitive is LocalV to the accuracy of the document indexing and retrieval steps?**
>
> We appreciate this question! We assessed LocalV’s sensitivity to the indexing and retrieval components through two key ablation studies:
>
> 1. Necessity. The retrieval step is essential: removing the index entirely led to a performance drop from 45% to 35%.
>
> 2. Robustness. The system is highly robust to the choice of backbone model. We substituted the primary planner with DeepSeek-V3 still achieved a comparable 41.6% pass rate. We attribute this robustness to our dual-level description strategy, which renders the indexing task well-defined and tractable. To further validate this, we manually annotated the golden document fragments that should be retrieved for the AES IP in RealBench and used them as a test set to evaluate retrieval quality. The results in the table below show that both LocalV (Claude) and LocalV (DeepSeek-V3) achieve high retrieval accuracy. This demonstrates that the indexing step is not a performance bottleneck, as our design supports consistent retrieval quality across models with different capabilities.
> | |Precision|Recall|
> |-|-|-|
> |LocalV (DeepSeek-V3)|0.93|0.93|
> |LocalV (Claude)|0.92|0.95|
>
> ---
> > **Q2. Can LocalV effectively scale to system-level designs exceeding 2,000 lines of RTL code or to multi-module integration scenarios?**
>
> We appreciate this question! LocalV is inherently designed for scalability because the most error-prone stage, **Coding**, operates on localized context, effectively decoupling generation difficulty from the overall system size. We address the reviewer's specific scenarios as follows:
> - **Multi-Module Integration:** LocalV supports this by treating submodule definitions (or existing RTL) as documentation references, enabling agents to integrate interfaces without requiring full implementation details.
> - **System-Level Scaling (>2,000 lines):** Our architecture is in principle capable of handling such scale, but we acknowledge a practical bottleneck in current foundation models: they require larger context windows to process long documents and maintain stable code output. At present, our method reliably produces correct code of approximately **600 lines**, which is already substantially larger than existing benchmarks such as RTLLM (average 56.1 lines) and VerilogEval (average 15.8 lines). Looking ahead, further optimizing context management within the agent should help bridge the gap toward generating even longer RTL designs.
>
> ---
> > **Q3. Will the authors release the LocalV implementation to facilitate reproducibility and community validation?**
>
> We appreciate this question! Yes, we plan to release the complete LocalV implementation on GitHub soon after the review period to facilitate community validation and future research.
>
> ---
> Thank you again for your valuable feedback. We hope the rebuttal is adequate in addressing your concerns, and if so, please update your review in accordance. If any questions remain, we welcome further discussion.

---

### Author Response · Authors · 2025-11-22
**Global response**

We thank the reviewers and ACs for their time and effort.

We are glad the reviewers acknowledge the following contributions:
- Our observation is insightful (D3Dj), and the idea is novel (aBM8) and principled (aBM8).
- Our method is coherent (aBM8, QRyn), technically grounded (aBM8), and effective (QRyn).
- The paper is clearly presented and logically organized (D3Dj, QRyn).
- The improvement is noticeable, and the component-level contributions are validated clearly (D3Dj).
- The paper contributes to automated hardware design by addressing the practical challenge of IP-level Verilog generation, and it is a meaningful step toward scalable and reliable RTL generation for practical IP-level design tasks (We sincerely appreciate QRyn's comment!).

The reviewers' main concerns focus on conducting further comparisons with previous work, providing additional validation of the information locality hypothesis and the retriever, offering deeper analysis of the debugging step, expanding benchmark evaluations, and assessing the quality of the generated circuits. To address these concerns, we have added comprehensive experiments, provided further explanations and analysis of our method, and revised the paper accordingly. We hope that these efforts resolve the reviewers' doubts.

**Summary of revisions**
- To provide deeper insights and identify directions for improvement (D3Dj, QRyn), we added **Appendix C** to investigate and categorize failure cases.
- To analyze the cost-benefit trade-off and per-iteration performance (aBM8, LYBq), we added **Appendix D**, which details the debug workflow (aBM8) and illustrates Pass@1 (Accuracy) vs. #iterations with token costs.
- To examine hardware efficiency metrics (LYBq), we added **Appendix E** to demonstrate the design quality of the code generated by LocalV.
- To better illustrate the concept for information locality (aBM8), we provided positive and negative examples in **Appendix F** to illustrate the characteristics for locality of documents.
- To strengthen the validation of the information locality hypothesis (aBM8), we conducted extended experiments as shown in **Table 1**.
- To further assess the generalization capabilities of LocalV(D3Dj, aBM8), we added an experiment on CVDP [1] benchmark in the **Experiments** section.
- We corrected typographical errors, such as the notation $\bar{H}_{\mathrm{norm}}$.

[1] Comprehensive Verilog Design Problems: A Next-Generation Benchmark Dataset for Evaluating Large Language Models and Agents on RTL Design and Verification

---

### Meta-Review · Area_Chair_aAk2 · 2026-01-06

**Summary:**

This paper proposed a multi-agent framework for IP-level Verilog (RTL) generation from natural-language specifications. The core insight is the Information Locality principle: in hardware design, each RTL semantic unit typically depends on a small, localized portion of the specification. Exploiting this property, LocalV decomposes the challenging long-document long-code generation problem into manageable, locality-aware subtasks.

**Reviewer Concerns:**

- Incremental Novelty vs. Prior Agent Systems:
Several reviewers questioned whether LocalV introduces fundamentally new agent coordination ideas or mainly refines existing multi-agent pipelines.

- Reliance on the Information Locality Assumption:
Concerns about how strongly this assumption holds across: different IPs, less structured or “flat” specifications and so on.

- Evaluation Scope and Generalization:
Initial reliance on RealBench alone was seen as narrow.
Reviewers asked for additional benchmarks, larger-scale designs, and multi-module/system-level validation.

One reviewer questioned whether unchanged PPA implies memorization of golden RTL or an “easy” benchmark.

**Reviewer Scores:**

D3Dj: likley maintain the score since novelty concern is only partially resolved
aBM8: unclear comments, ignored.
LYBq: score likely unchange or even descrese because of the PPA concern, which seems to be strange to me too.
QRyn: score to be maintained or slightly raised.

I think this paper sits at the boarderline. I am slightly concerned by the unchanged PPA. Although I liked the key observation that the authors made on dealing with the long-context in the verilog generation problem. I think
1. the IP-level geneartion is a bit too specific
2. the correct audiences of this paper may be in the EDA community - the ML side of the novelty of this paper is fairly limited.

---

### Decision · Program_Chairs · 2026-01-26

Reject